# TiO_2_ as an Anode of High-Performance Lithium-Ion Batteries: A Comprehensive Review towards Practical Application

**DOI:** 10.3390/nano12122034

**Published:** 2022-06-13

**Authors:** Sourav Paul, Md. Arafat Rahman, Sazzad Bin Sharif, Jin-Hyuk Kim, Safina-E-Tahura Siddiqui, Md. Abu Mowazzem Hossain

**Affiliations:** 1Department of Mechanical Engineering, Chittagong University of Engineering and Technology, Chittagong 4349, Bangladesh; sourav@cuet.ac.bd (S.P.); safina@cuet.ac.bd (S.-E.-T.S.); mowazzem@cuet.ac.bd (M.A.M.H.); 2Department of Mechanical Engineering, International University of Business Agriculture and Technology, Dhaka 1230, Bangladesh; sazzad.sharif@iubat.edu; 3Clean Energy R&D Department, Korea Institute of Industrial Technology, 89 Yangdaegiro-gil, Ipjang-myeon, Seobuk-gu, Cheonan-si 31056, Chungcheongnam-do, Korea

**Keywords:** lithium-ion battery, electrochemical anodization, titanium dioxide, nanostructure

## Abstract

Lithium-ion batteries (LIBs) are undeniably the most promising system for storing electric energy for both portable and stationary devices. A wide range of materials for anodes is being investigated to mitigate the issues with conventional graphite anodes. Among them, TiO_2_ has attracted extensive focus as an anode candidate due to its green technology, low volume fluctuations (<4%), safety, and durability. In this review, the fabrication of different TiO_2_ nanostructures along with their electrochemical performance are presented. Different nanostructured TiO_2_ materials including 0D, 1D, 2D, and 3D are thoroughly discussed as well. More precisely, the breakthroughs and recent developments in different anodic oxidation processes have been explored to identify in detail the effects of anodization parameters on nanostructure morphology. Clear guidelines on the interconnected nature of electrochemical behaviors, nanostructure morphology, and tunable anodic constraints are provided in this review.

## 1. Introduction

Lithium-ion batteries (LIBs) have made undeniable contributions to modern energy technology [1], especially in the era of portable devices such as laptops, phones, and cameras. Its lightweight nature has made this type of battery the first choice for pure electric vehicles (PEV), plug-in hybrid vehicles (PHEV), and hybrid vehicles (HEV). Higher energy density both per unit weight and per unit volume have made LIBs lightweight. Moreover, their long useful life, green technology, easier manufacturing process, low production cost, and lack of memory effect are the other sources of attraction for LIBs [2,3,4,5,6,7,8]. M. Stanley Whittingham, while working for the oil company Exxon, first proposed the idea of rechargeable LIBs in the 1970s. The LIB was composed of metallic lithium as an anode and layered titanium disulfide as the cathode. Later, in the early 1980s, Goodenough enhanced the rechargeable LIBs with a layer of oxide cathode materials [9]. Then, in 1991, Sony commercially introduced LIBs in the market, where graphite had been used as the anode material due to its availability and its theoretical specific capacity of 372 mA hg^−1^ [10]. However, LIBs have enough positive characteristics to be one of the feasible energy storage systems for meeting today’s increasing demands for efficient energy. At present, extensive applications of LIBs are physically observed in robotics, uninterrupted energy supply (UES), different power tools, and stationary power storage units. In addition, vehicles for long-range driving require at least five to ten times more specific energy density than that of LIBs (150 Wh kg^−1^) [11]. Figure 1 displays the growing dependency and consumption loads on LIBs over time. The trend shows that the consumption of LIBs for all kinds of EVs has increased exponentially, and consumer consumption has greatly increased [12,13]. Conventional graphite material has some severe drawbacks. Solid electrolyte interface (SEI) formation is a common issue that limits the capacity, resulting in rapid capacity fade. Moreover, the low working voltage of ~0.1 V vs. Li^+^/Li makes possible applications such as lithium electroplating. In addition, dendrite formation due to high polarization at high charging/discharging rates causes short circuiting, cell damage, and overheating during operation, which pose safety issues. It is noted that its application is limited to low operating temperature (−30 °C to 60 °C), which does not allow for applications at high operating temperatures such as in EVs and HEV, and that the low operation temperature tends to facilitate the dendrite formation [2,14,15,16,17,18]. Therefore, advanced technological modifications of the present conventional materials used in batteries have to be introduced for far better energy efficiency.

Many materials with potential like transition metal oxides (TMO, comprising TM = Ti, Ce, V, Ni, Co, etc.), layered TMOs (Cr, Ni, Fe, V, etc. and their composites) [19], silicon-based materials (Si/C, Ni_3_Si_2_/Si, and Si nanostructures; Cu-Si-Al_2_O_3_, etc.) [20]; carbon-based materials; and lithium-based metal alloy Li_x_M (where M = Sn, Al, Ge, Gb, Pb, Sb) have emerged as anode candidates [21]. According to the lithium insertion/deinsertion mechanism, these anodes are stratified into three groups: (i) alloy anodes like Sb, Sn, Si, Pb, Ge, (ii) conversion anode (TMOs, metal sulfides, nitrides, phosphides, etc.), and (iii) intercalation anodes such as carbon-based material, lithium-based composites, and Ti-based oxides [22].

The reversible capacities of different anode materials are shown in Figure 2, where it is observed that TiO_2_ is an intercalation anode from the TMO group that possesses relatively few reversible capacities compared with others. However, it can be a trendsetter due to its higher Coulombic efficiency and cycle stability. Moreover, overcoming the disadvantages of graphite, TiO_2_ can also be a pioneering anode for LIBs due to its low volume fluctuations (<4%) during intercalation, faster kinetics for lithium-ion insertion/extraction, and good capacity retention. In addition, the high working potential of 1.7 V vs. Li/Li^+^ ensures safety by suppressing dendrite formation and provides the low polarization effects that are extremely desired for high-performance LIBs. Furthermore, the less toxic effluent produced by TiO_2_ has emerged as a promising anode candidate for LIBs, but the low electron conductivity and slow ion diffusivity hinder its application. However, the ion diffusivity depends on the particle size, which can be calculated with the following Equation (1):ƛ = L_ion_^2^/D_Li_(1)
where ƛ is the diffusion time, L_ion_ is the diffusion length, and D_Li_ is the particle size of Li^+^ [23]. This equation implies that faster diffusion can be achieved by size tailoring into nanoscale. Reducing the particle size into the nanoscale showed the immediate improvement of the diffusion kinetics as well as of rate performance, as the diffusion lengths are shortened. However, the specific surface area increases more than the bulk, which confirms more electrodes and electrolyte interfaces [24]. In addition, 3D morphology with a hierarchal structure, especially the highly ordered nanotubular array with optimized aspect ratio, plays a crucial role in battery performance as the electrolytes favor the electron and lithium transport properties in the electrode [25,26].

## 2. Electrochemistry of LIBs

LIBs convert chemical energy to electrical energy through electrochemical reactions (oxidation/reduction) and vice versa. It is noted that LIBs comprise three main components: a positive electrode (cathode), a negative electrode (anode), and an electrolyte. A separator isolates the anode and the cathode to prevent short circuit. The electrolyte is employed for carrying ions during the redox reaction, and the electrons flow through the outer circuit. During discharging (Figure 3), oxidation occurs at the anode, producing positively charged Li^+^ ions and negatively charged electrons. The Li^+^ and the electrons flow through the electrolytes and the outer circuit, respectively, serving as transportation media but not taking part in reactions. In contrast, the cathode reduces to accommodate the Li^+^ ions and electrons. During charging (Figure 3), the flow of electrons and Li^+^ ions is in the opposite direction, where Li^+^ now moves from positive to negative electrodes. The insertion and extraction of Li^+^ ions from the electrodes are known as intercalation and de-intercalation, respectively [27,28,29]. 

Figure 4 shows the supremacy of LIBs over other rechargeable batteries with respect to energy density (Wh kg^−1^). However, LIBs also face some challenges, for instance in their performance determinant criteria: potential, specific capacity, energy density, etc. The cycle life and lifespan are reliant on the nature of the electrode and electrolyte interfaces, as well as the stability of the electrode materials and their interfaces [30]. The performance efficiency of LIBs depends heavily on the inherent properties of their anode materials. Hence, the development of the current situation in regard to LIBs is one of the major focuses of extensive research to develop an appropriate anode for distinctive application purposes. To fulfill the criteria for an ideal anode, LIBs require (a) highly exposed energy facets to provide more Li^+^ sites, (b) higher specific surface area to ensure more interfacial contact between electrolyte and electrode surface, (c) low volume fluctuation during lithium intercalation/de-intercalation, (d) high electronic conductivity to ensure fast charging and discharging rate, (e) higher lithium diffusion rate confirmed by short diffusion length and larger pore size, (f) low intercalation potential vs. Li^+^/Li, and (g) green technology [16,31].

## 3. Different TiO_2_ Polymorphs

The promising Li^+^ insertion capacity of TiO_2_ has drawn the attention of researchers to advance the research on using this material as anodes for LIBs. The Li^+^ insertion/extraction mechanism takes place via Equation (2):Li^+^ + TiO_2_ + xe^−^ ↔ Li_x_TiO_2_ (0 ≤ x (mole fraction) ≤ 1)(2)

Bulk TiO_2_ exhibits a high theoretical capacity of 335 mA hg^−1^ compared with that of lithium titanate spinel, 175 mA hg^−1^. Extensive studies on TiO_2_ find that this material appears with eight different crystal structures: anatase, rutile, brookite, TiO_2_-B (bronze), TiO_2_-II (columbite), TiO_2_-H (hollandite), TiO_2_-III (badelleyite), and TiO_2_-R (ramsdellite). Among all the polymorphs, anatase, rutile, brookite, and TiO_2_-B are investigated to a great extent as they are more chemically reactive than others [6]. Under density functional theory, the crystal structures’ stability descends in the following order: anatase, rutile, brookite, ramsdellite, spinel [32].

Anatase is a meta-stable polymorph and is the most suitable for lithium insertion/deinsertion among all the crystal structures of TiO_2_. This tetragonal crystal structure from symmetrical space group *I4_1_/amd* possesses a body-centered texture and appears with a distorted c-length. Each octahedral TiO_6_ site shares edges including corners with two other adjacent octahedra, as shown in Figure 5a [33]. The channels created among these octahedral sites make easier pathways for Li^+^ insertion. During intercalation, the unit cells lose symmetry, and with the increase in the lithium inserted, the phase transformation starts. It is reported that at lithium mole fraction x = 0.5, that is, Li_0.5_TiO_2_, the phase completely transforms from tetragonal into orthorhombic from space group *Pmn2_1_* [34,35]. The b-length expands, the c-length shrinks, and volume change occurs at ~4% with rapid capacity degradation [36]. Bulk anatase can accommodate a maximum of 0.5 mole Li^+^ ions per 1 mole TiO_2_ [37]. Flat charge–discharge plateaus indicate the biphasic transition of bulk anatase. Splitting the bulk into nanoscale <100 nm appears to give propitious results with shortening the plateau region, indicating divergence from the equilibrium biphasic transition and acting like a solid solution [38]. The reduction in size improves the intercalation mechanism and the charge/discharge capacities over 0.5Li and reported a maximum of 1 Li for 1 TiO_2_ [39,40].

In addition, rutile is the most stable crystal structure among all the crystal structures of TiO_2_ [41]. It is noted that of all the polymorphs, rutile also comprises octahedra of Ti cations surrounded by six-fold oxygen anions [42]. Its crystal structure forms in a symmetric tetragonal structure with space group *P4_2_/mnm*. The TiO_2_ octahedra share edges in the c-direction and corners in the ab plane as shown in Figure 5b [43]. The short c-length differentiates the structure from anatase, and the corners of the octahedra are placed on the ab planes [44]. In addition, lithium ions intercalated in the structure are found near the ab plane tetrahedral sites. This implies the difficulties of lithium insertion with a small diffusion coefficient, 10^−14^ cm^2^ s^−1^. Finally, the null intercalation of lithium is reported with the time being due to the Li-Li repulsive action [45,46,47]. Nanosizing interestingly increases the capacity of the rutile structure, showing particles with nanosized capacities of 0.23 mol to 0.8 mol per TiO_2_ and 0.1–0.25 mol/mole TiO_2_ for micro-sized particles [48,49].

TiO_2_-B is a monoclinic crystal structure from space group *C2/m* where TiO_6_ polygons with wavy edges are shown (Figure 5c). They are connected via corners. The density is minimal, which allows for a simple lithium pathway. The void structures compensate for the volume deformation during the lithium insertion without lattice deformation. The larger Ti-O bonds (2.20–2.25 Å) transform the structures of the octahedra into square pyramids [48,50]. This lithium insertion mechanism is not based on diffusion kinetics as with anatase and rutile; rather, it is controlled by a faradic pseudo-capacitive process. The insertion capacity appears to show promise as it closely resembles anatase. 

In addition, an orthorhombic variant of TiO_2_ is brookite from space group *Pbcv*. It possesses larger cell volume with larger density than anatase and rutile. This metastable crystal structure shares three edges (Figure 5d) of TiO_6_ octahedra with adjacent octahedra. This compact texture makes Li^+^ ion insertion more difficult. Recent studies have shown that modified brookite rather than bulk showed enhanced electrochemical properties that are now attracting the attention of more extensive research. The stability of brookite increases when the bulk is treated to nano scale, resulting in lower surface energy. Particle sizes ranged from 20 to 50 nm and showed a maximum reversible specific capacity of 60 mA hg^−1^ after 50 cycles. Modifying the material with a conducting material like brookite–TiO_2_ carbon nanotubes showed improved reversible capacity of 160 mA hg^−1^ after 50 cycles, which corresponds to less than 0.5 mol Li per TiO_2_ [51].

### Influence of Experimental Conditions on TiO_2_ Crystal Structures

The degree of crystallinity depends on its preparation techniques and related conditions such as pH, annealing temperature, and methods. Many methods have been employed for forming TiO_2_ crystal structures. The effects of pH are crucial for optimizing the crystal lattice. Ibrahim et al. [52] showed that higher crystallinity is promoted by lower acidic environments and that anatase can form at pH 9. A simple sol-gel technique was applied to TiO_2_ nanocrystalline powders of 3–7 nm particle size, and when pH 1 was applied, pH increased from 2 to 7, and a mixture appeared in anatase and rutile phase. After a further increment, the mixture turned into brookite [53]. Molea et al. [54] produced a nanocrystal powder of TiO_2_ with varying crystal structures through the hydrolysis method. The low acidity of pH 3 allowed for a mixed phase of anatase (~30%) and rutile (~70%), and increasing pH showed phase transformations. At pH 8.5 and 10, a pure anatase crystal phase was observed with different lattice parameters. Yang et al. [55] synthesized TiO_2_ nanorods with exposed crystal facets via hydrothermal technique and supported the argument of a pH–crystal structure relationship. The results indicated that only brookite phase appeared below pH 8 and that pH 8–10 facilitated a mixed rutile-brookite phase. A pure anatase phase was simultaneously observed for pH more than 10. In addition, crystallography is significantly related to annealing temperature. Barki et al. [56] reported that anatase phase formation started at 300 °C and that the anatase-to-rutile phase was established at 900 °C, as shown in Figure 6. It is noted that the anatase-to-rutile phase transformation occurs at 600 °C. Amorphous TiO_2_ nanotube transfer into the fully developed anatase phase was exhibited in XRD analysis when calcined at 500 °C. Calcination prolonged to 850 °C showed intense peaks of rutile increase and parallel decreases in the anatase nanotubes, as shown in Figure 6 [57].

## 4. Different Nanostructures

Nanostructured materials have some outstanding beneficial features, including more adaptable space for easier rebuilding, since their nano size increases the capacity regarding confinement effect, better mechanical stability, and greater surface area, all of which make them excellent for photocatalytic activities. The nano scaling of materials has been extended through different dimensions such as nanopores, nanoneedles, nanorods, nanotubes, nanoparticles, nanogravels, nanowalls, nanoarrays, nanocones, hollow microspheres, nanocomposites, and nanofilms [58]. These advantageous nanofeatures can be applied in gas sensors [59] and structural building materials [60]. The different nanostructures of TiO_2_ with various morphologies exhibit distinguishing properties. Among them, granular TiO_2_ is more developed and unique in that it has a larger surface area and a larger boarded band gap compared with bulk TiO_2_, which provides ease of electron transport and more interfacial interaction between electrolytes and electrodes (Figure 7). Hence, they facilitate improved electrochemical characteristics [61].

### 4.1. One-Dimensional Structure

In recent years, 1D nanostructured TiO_2_ has attracted a great deal of research attention including as nanowires, nanorods, nanoneedles, nanobelts, and nanotubes. They have been extensively investigated as they possess both all the properties of nanoparticles and some unique properties as well. These nanostructured materials are fabricated with at least one dimension ranging from 1 to 100 nm and have high aspect ratios [62]. Nanobelts are one of the ideal 1D nanostructures of TiO_2_. First, the higher crystallinity of TiO_2_ nanobelts possesses a comparatively high specific surface area. Second, it allows electron pathways in the axial direction. Moreover, the synthesizing process is easier [62,63]. 

Tian et al. [63] reported a belt-like special heterostructure of TiO_2_ that could be designed by assembling secondary nanoparticles with special morphologies such as the recombinant features of nanowires, nanorods, nanoneedles, and nanotubes on the surface of 1D nanostructures. Different strategies have been applied for synthesizing metal nanoparticles/1D nanostructures including wet in situ reduction, photo reduction in situ, and immobilization. The output products range from 3–5 nm in diameter and are more uniformly distributed on the surfaces of 1D nanobelts. Moreover, as no inorganic reagent is used in this process, it is more eco-compatible and nontoxic. In addition, the immobilization method has been introduced in many research works where metal nanoparticles or 1D nanobelt surfaces are immobilized or both are immobilized and joined to make an interconnection. This is advantageous, with precise control over the metal nanoparticles as well as the morphology, but the modification of surfaces is essential with this method [64,65,66]. Furthermore, the outcomes of this method cannot truly be justified as heterogeneous aggregation as the nanoparticles are connected with the nanostructure substrate by chemical bonding. In in situ methods, the lattices are connected with the substrate like a physical attachment [67]. 

One nanomaterial, 1D nanobelts, has drawn the considerable attention of research groups as negative electrodes for LIBs. As Li^+^ has a relatively large specific area, which allows for more interfacial reactions between electrode and electrolyte, more Li^+^ intercalation is possible. In addition, its high degree of crystallinity and porous structure facilitate more electron conductivity and diffusion. Furthermore, the layered structure of nanobelts ensures more stable and flexible architecture during Li^+^ insertion/extraction. However, as an energy storage system, it is inadequate because of its inherently low electronic conductivity. This is attributed to its large surface area, which causes low current density and consequently increases the charge/discharge rates. The large surface area causes side reactions with electrolytes, and thus, the solid electrolyte interface gets thicker. This thicker layer impedes the electron and Li^+^ ion transport. Furthermore, the high charge/discharge rate is not always beneficial as the aggregation of nanoparticles causes inter particle resistance. Therefore, to overcome this difficulty mixing the nanobelts with conductive materials such as C/TiO_2_ nanowires, C/TiO_2_ nanotubes, functionalized graphene sheet/TiO_2_ nanorods, and Sn/TiO_2_ nanotubes can be an effective solution [68].

It is noted that nanostructured TiO_2_ has better electrochemical activity than bulk TiO_2_. Many research studies reported that among all the polymorph structures of TiO_2_, the anatase crystal structure has shown better electrochemical performance as an anode of LIBs by providing better Li^+^ ion storage capacity. The ideal discharge process of anatase TiO_2_ can be classified into three steps. In the first step, the monotonous potential decrease is happened by forming conductive Li_x_TiO_2_ in the solid solution stage. Phase switching occurs at the second stage via the coexistence of Li_poor_ phase (tetragonal Li_0.001_TiO_2_) with Li_rich_ phase (orthorhombic Li_0.5_TiO_2_) at equilibrium, which represents the reversible insertion and extraction of Li^+^ ion from and to the octahedral sites of anatase TiO_2_. In the third step, all Li^+^ ions are stored on the surface of the anode by creating a phase Li_1_TiO_2_ where intercalation potential x = 1. This is attributed to the pseudo-capacitance of the anode material. This pseudo-capacitance behavior is influenced by the morphology and crystal structure of the TiO_2_ anode material. Therefore, improving the surface-to-volume ratio can increase the pseudo-capacitance property along with the interfacial contact between electrode and electrolyte as well as shorten the Li^+^ ion diffusion rate. Hence, the lithium storage capacity is increased. However, some serious shortcomings limit its advantageousness like low thermodynamic stability and difficult handling. To solve these problems, a micro-hierarchal nanostructure can be a feasible choice, as it can provide micro-hierarchal building blocks, high current density, and thermodynamic stability. Mesoporous TiO_2_ composed of nanocrystals has attracted much attraction in this regard as it not only possesses a nano/micro hierarchal structure but also is blessed with high surface area and high pore volume [16,68].

Tian et al. [69] introduced a simple method for synthesizing one-dimension mesoporous anatase TiO_2_ (1DMA-TiO_2_) where the output product comprises both a stable hierarchal structure and all the unique properties of one-dimensional nanoparticles. In the synthesis, 1D TiO_2_ provides an efficient pathway for electrons and shorter diffusion length for ions. A two-step fabrication method was followed where carbon nano tubing (CNT) was used as a template. Morphological studies of anatase nanoparticles with an average size of 5–12 nm give 96.3% Columbic efficiency, longer cycle life, high-rate capability, and high reversible capacity, as depicted in Figure 8. This positive performance can be attributed to the peculiar architecture of 1DMA TiO_2_, which is facilitated not only by its nano/micro hierarchal structure but also by its simple fabrication, thereby providing an efficient electron pathway. The improved surface area ensures both the larger interfacial contact and the better thermodynamic stability due to agglomeration and strain upon cycling. However, research can be extended in this regard by altering some synthesizing parameters such as template and parent materials to further the improvement of the surface area, which was visible in conventional anatase mesoporous TiO_2_ with a specific surface area near 505 m^2^ g^−1^. This would be helpful in contributing to understanding the electrochemical behavior of 1DMA TiO_2_.

Tang et al. [70] synthesized a hybrid structure, TiO_2_ nanotubes on graphene sheets (Gr-TNT) as shown in Figure 9a–d, through a combination of graphene and TiO_2_ nanotubes to improve the electrochemical behavior of the material. As graphene is highly electroconductive and will manifest a highly charged diffusion rate along with more Li^+^ ion storage on the composite structure surface, many research efforts have been applied to developing composites such as nanostructures of TiO_2_ (including nanoparticle, nanorods, nanoneedles, nanosheets, etc.). Here, a mild hydrothermal process is applied to prepare TiO_2_ nanotubes on graphene nanosheets. TiO_2_ colloids were used as seeds on the graphene oxide (GO) sheets, and the sheets were used as the plane of the center of the nucleation of nanotubes. Gr-TNT as a preferable anode material for LIBs exhibited better rate performance, better cycle stability, and more Li^+^ storage capacity than not only any other similar nanostructure like Gr-TiO_2_ but also than bare TiO_2_ (80 mA hg^−1^ at 10C). Hence, the combined effect of high conductive graphene sheet/TiO_2_ nanostructure improves electron conductivity, accommodates the strain due to intercalation/deintercalation, and gives more surface area, allowing more interface between the electrodes and the electrolyte; the hierarchal architecture is credited to these improvements.

Tammawat et al. [71] introduced another unique 1D nanostructure of TiO_2_ called nanofibers that is drawing considerable attention as it can be applied as an anode material for LIBs without mixing it with any binder or conductive material. Electrospinning was deployed to fabricate the nanofibers followed by calcination at 400 °C, 600 °C, 800 °C, and 1000 °C for 3 hrs. The average diameter of the prepared nanofibers was 200 nm, and that for the for calcined nanofibers at 400–800 °C were 100–120 nm. This is attributed to the contamination of the carbon-based amorphous polymer. After calcination, this content reduced to a different weight percentile corresponding to the calcination temperatures and reported 0.35, 0.09, 0.14, and 0.74 weight percent (wt. %) for 400–1000 °C. Moreover, 1000 °C calcined products were found to be interlinked particles of size 100 nm. Better electrochemical performance was observed for anatase, as it is known that electrochemical performance increases with decreases in crystallite size. The Columbic efficiency ranged from 96% to 100%. The positive outcomes for 400 °C calcined nanofibers could be attributed to the high surface-to-volume ratio, higher nonstoichiometric parameters (Li^+^_rich_ and Li^+^_poor_), and high electron conductivity ensured by 100% active material.

Nanowires were prepared from TiO_2_-B polymorphs by Armstrong et al. [72] following a simple preparation technique. The stable architecture of the prepared nanowires facilitates capacity retention, and for 2 to 100 cycles (Figure 10a), the capacity-fading gradient was 0.1% at 50 mA g^−1^ (C/4 rate); even better, it was 0.06% for 20–100 cycles. High rate capabilities as shown in Figure 10b were available, such as 160 mA hg^−1^ at current density 500 mA g^−1^ and 100 mA hg^−1^ at 2000 mA g^−1^. Furthermore, the potential of about 1.6 V vs. Li^+^ (1 M)/Li ensures more safety than many other anodes like alloys. All these good features make the TiO_2_-B nanowires promising candidates for LIBs, though more focus should be given to understanding and disseminating the exact origin of irreversibility. Conductive coating over the materials is a possible solution that can considerably increase the electronic conductivity.

TiO_2_ nanorods are 1D structures. Floating and freestanding TiO_2_ nanorod films were prepared by Xia et al. [73] at the liquid–gas interface through a simple hydrothermal process and used as additive-free anodes of LIBs. As an anode of LIBs, this material showed exceptional qualities. First, it can be used as a conductive agent-free and binder-free anode. It exhibited a first-cycle discharge capacity of 31.44 mA hg^−1^ at 30 mA g^−1^ current density. After 100 cycles, the capacity was 32.66 mA hg^−1^, which is larger than the initial capacities shown in Figure 11a,b. This result can be explained as the chemical stability and simpler electron pathway provided by the structure. However, the specific capacity was too low compared with the other structures of TiO_2_. Hence, it cannot serve as a fit anode for LIBs. Making a hetero nanostructure with this material and coating with conductive material like carbon can be fruitful for improving specific capacity, charge/discharge rate, rate capabilities, and cycle stability.

Nanocomposites of hexagonal MoO_3_ inlaid with highly crystalline TiO_2_ nanoparticles have been synthesized by Adam Kubiak et al. [74] following a template-assisted microwave method. The optimum molar ratio of 5:5 was found to give the best outcomes. That nanocomposite as an anode of LIBs could sustain 100 cycles, with a reversible discharge capacity of 700 mA hg^−1^, which is more than two times that of anatase TiO_2_. The highly active electrochemical nature of MoO_3_ is the precursor to these outstanding outcomes, and anatase TiO_2_ nanoparticles also have great impacts through making the cyclic operation safer, maintaining the stability of the architecture, and suppressing the volume fluctuations.

Another TiO_2_ nanocomposite comprising nanocrystal TiO_2_ and N-doped carbon nanotubes (TiO_2_@C/N) was used as binder-free anode material for LIBs. The anode also showed very beneficial results. It showed 1054.7 mA hg^−1^ initial discharge capacity, much higher than the anatase TiO_2_ nanocrystal. The synergistic effects of highly conductive N-doped carbon nanofibers and highly porous TiO_2_ nanocrystals are the contributing factors to the superior cycle stability, high rate capability, and high reversible capacities. Porous TiO_2_ nanocrystals embedded on carbon nanotubes offered simple channels for charge transfer and shortened the diffusion lengths. Moreover, integrated flexible mechanical structure was offered by N-doped carbon nanotubes [75].

One-dimensional nanoribbons showed higher electrochemical behaviors as anode when compiled as a nanocomposite with SnO_2_. A reversible discharge capacity of 265 mA hg^−1^ was displayed. The main drawback of SnO_2_ is rapid capacity degradation due to architectural instability and high-volume fluctuations. When TiO_2_-(B) nanoribbons are implanted, the mechanical supports are ensured, and at the same time, higher specific capacity is maintained by SnO_2_ [76].

### 4.2. Two-Dimensional Structures

Large specific surface area, highly exposed facets, and short diffusion lengths are very necessary for high Li^+^ storage, and 2D (two-dimensional) frameworks of a material can facilitate these properties. Moreover, open charge transport and the stability of architecture during Li^+^ intercalation/de-intercalation are also convenient for the same purpose. As nanostructured TiO_2_ has been established as a promising anode material for LIBs, more significant investigations have attempted to make it more operational by introducing many 2D nanostructured frameworks of TiO_2_ including nanosheets, nanoflakes, nanobelts, and nano hollow spheres.

Chen et al. [77] introduced ultrathin carbon-supported anatase TiO_2_ nanosheets (C-TiO_2_ NS) following a simple two-step solvothermal technique. Due to the high electronic conductivity of carbon, the sample is seen to possess better electrochemical performance than antase-TiO_2_. It exhibited high specific charge/discharge capacities of 203 mA hg^−1^ and 314 mA hg^−1^. At 1C, ~100% of the theoretical capacity of anatase TiO_2_ held at 170 mA hg^−1^, while at a high current rate of 5C, it lessened to 88% of the theoretical capacity even after 100 cycles, which is still very promising. Thus, better rate capability and high cycle efficiency were the reasons for the highly stable microstructural architecture facilitated with void space and an exposed (001) energy facet. J. S. Chen [78] in another work produced a pure anatase TiO_2_ nanosheet through a simple hydrothermal process. Morphological investigations through FESEM and TEM suggested that the prepared nanosheets were squares of the dimensions 20–100 nm and thickness of 10 nm. These dimensions are advantageous as they make the structure robust in volume change during Li^+^ insertion/extraction. In addition, these average dimensions contributed to exposing 62% of the high energy facets (001) (Figure 12).

A scalable template-assisted sol-gel method has been applied to synthesize 2D TiO_2_ mesoporous nanosheets. The interfacial stain between the template KCl and TiO_2_ suppressed the crystallization process of TiO_2_, though the reaction temperature was maintained about 400 °C. The amorphous TiO_2_ nanosheets possessed high pseudo capacity, improved diffusion kinetics, and high durability. As a result, it exhibited 103 mAhg^−1^ reversible capacity at high C rate 6 Ah g^−1^ even after 1000 cycles [79].

Carbon-coated TiO_2_ nanosheets/reduced graphene oxide were prepared by Shang Jiang et al. through a simple one-pot solvothermal process whereby 10 nm ultrathin nanosheets are homogeneously moored on the reduced graphene sheets. This heterogenous hierarchical structure provides simple pathways for lithium charge transfer and higher cycling stability. As an anode, this nanocomposite displayed 401 mA hg^−1^ after 200 cycles at a current density of 100 mA g^−1^ and 126.5 mA hg^−1^ at a current density of 2500 mA g^−1^. These results confirm better rate capability and cyclability than pure anatase TiO_2_ nanosheets [80].

Qinghua Tian et al. used 2D nanosheets of TiO_2_ to make composites with 0D SnO_2_ nanoparticles [81]. A simple morphology-maintained phase transformation process was applied. Enhanced lithium storage capability was indicated through high specific capacity of 758 and 474 mA hg^−1^ at 200 after 390 and 650 cycles, respectively, and even 1000 mA g^−1^. The combined effects of the outer carbon layer and the inner robust structure of the 2D TiO_2_ nanosheets and the 0D SnO_2_ nanoparticles are the main attributes for better electrochemical kinetics, higher architectural stability, and enhanced lithium-ion storage capacity.

Dendrite formation is a serious issue in battery cycling. Moreover, high volume fluctuations cause rapid capacity decline. To suppress these issues, another approach has been adopted involving synthesizing a novel 2D TiO_2_ nanosheet with graphene that is then coated with 2D black phosphorous nanosheets (BPN). This heterostructure promotes rapid interlayer charge transfer paths and higher lithium storage capability. Both BPN and TiO_2_ nanosheets possess a highly conductive nature due to their advantageous nanoarchitecture. In consequence, superior rate capability (271.1 mA hg^−1^ at 5 A g^−1^), enhanced cyclability (502 mA hg^−1^ after 180 cycles), and highly improved storage capacity (initial discharge 1336.1 mA hg^−1^ at 0.2 A g^−1^) were observed within the 1–3 V potential window as anodes. These outcomes are far better than any pure nanostructured anatase TiO_2_ [82].

Jorge Lopez et al. [83] prepared a nanocomposite of TiO_2_/C where unique 2D nanoparticles were used with 2D carbon nanofibers. The heterogeneous nanostructure showed improved rate capability (200 mA hg^−1^ at 500 mA g^−1^ current density), good capacity retention (~99%), high initial discharge capacity (683 mA hg^−1^), and better cycle stability (290 mA hg^−1^ after 100 cycles at 100 mA hg^−1^ current density). The layered TiO_2_ nanoparticles allowed for fast charge transfer and improved the diffusion kinetics. Moreover, the highly conductive 2D carbon nanofibers accelerated the ion conductivity and resulted in higher electrochemical performance.

Huang [84] introduced a new 2D hybrid nano-structure G-TiO_2_(B) from the synergistic effect of a 1D mesoporous TiO_2_-B nanobelt and a conductive 2D graphene sheet as a high-power anode for LIBs. The preparation was followed by a simple hydrothermal method in the presence of graphene sheet and calcined at 400 °C for 4 h. This hybrid nanostructure is theoretically converging to high electrochemical behavior such as that for possessing 1D mesopores. In addition, the structure facilitates short diffusion lengths for Li^+^ transport, ensures high interfacial contact between the anode and electrolytes, and provides simple electron pathways. Here, a graphene sheet, due to its high electron conductivity, can serve as a superior current collector with high specific capacities and rate performance. 

Another hetero nanostructure that combined ultrathin 2D mesoporous TiO_2_ and reduced graphene showed very good cyclability and rate capability (245 mA hg^−1^ at 1 A g^−1^ after 1000 cycles). The volume fluctuations during lithium insertion and extraction were very slight due to the flexible mechanical support from 2D mesoporous TiO_2_. In addition, faster kinetics and short diffusion rate of ions were confirmed by this stacked layer structure [85].

Researchers developed another material called r-GO@TiO_2_(B)@Mn_3_O_4_, a multifunctional, multidimensionally ordered yolk-membrane-shell hybrid nanostructure in which TiO_2_(B) nanosheets acted as major active materials. Most the capacity contributions came from TiO_2_(B) as well as the safety and durability. Meanwhile, the auxiliary active material Mn_3_O_4_ possessed a high initial discharge capacity of 935 mA hg^−1^. The r-GO nanosheets acted like an ion host and absorbed the mechanical stress of TiO_2_(B). The aggregation of Mn_3_O_4_ was suppressed by TiO_2_(B), which in consequence caused strong lithium storage and delivering capacity (662 mA hg^−1^ at 500 mA g^−1^ after 500 charge discharge cycles) [86].

TiO_2_ hollow spheres with highly exposed (001) energy facets were synthesized by S. Ding et al. [87]. The microspheres were composed of ultrathin nanosheets. A simple hydrothermal preparation technique was followed, and the prepared samples were calcined at 550 °C for 8 h, confirming the anatase phase. The initial irreversible capacity loss (~32%) and comparatively lower reversible capacities formed a solid electrolyte interface (SEI) layer along distorted Li^+^ insertion sites. Again, the voltage plateaus 1.7 V (cathodic sweep) and 2.1 V (anodic sweep) create a polarization effect greater than that of many nanosheet TiO_2_ structures. The origins of these shortcomings need further consideration for far better results.

Wu et al. [88] synthesized a petal-like TiO_2_ nanosheet through a simple and green preparation technique. The first discharge/charge capacities were reported as 382 mA hg^−1^ and 326 mA hg^−1^ at low current density (20 mA g^−1^) with an initial irreversible capacity loss of 14.7%. The loss decreased to null for the subsequent cycling. The capacities are low compared with other Ti-based anode materials, which can be attributed to the adsorption of trace water by the high surface area during preparation. At high current density (400 mA g^−1^), the discharge capacity after 50 cycles can maintain 93% of the initial capacity. These values suggest the good cycle stability of the prepared samples. Regarding the rate capabilities, the anode material exerted moderate behavior, but to gain deeper insights into rate performance, more experiments with high rates should be taken under consideration.

Sandwich-like stacked structures of carbon-supported ultrathin titanate nanosheets (CTN) are a new dimension in the 2D nanostructure group of titanium-based anode materials. Liu et al. [89] introduced a new simple method of fabricating such a wood-like structure through a hydrothermal technique. The Oswalt ripening effect limits the diffusion at a certain point. Herein, the author proposed an ion exchange liquid (IL) as a group of solvents to overcome these challenges. IL possesses some noble properties like low volatility, high boiling temperature, high design ability, and high solubility. Regarding the electrochemical performance, first, no plateaus for certain charge/discharge voltage were found. These continuous gradient curves for charge/discharge capacity can be attributed to the unique structure of CTNs because the strong interaction between amorphous carbon and ultrathin nanosheets of TiO_2_ improves the grain boundaries. More precisely, the Li^+^ ion diffusion and charge transfer take place at the surface of CTNs like electrochemical capacitors, and thus, no voltage plateaus were observed clearly. A high-rate capability of 190 mA hg^−1^ at 50C was noted to be an output function with a unique sandwich-like structure and the superior conductivity of carbon nanoparticles.

Chen et al. [90] prepared another sandwich-like nanostructure wherein carbon-coated ultrathin TiO_2_ nanosheets (~14 nm) were endowed with defect-rich MoS_2_ from both sides via intimate interfacial contact. The defect-rich few layers of MoS_2_ and the ultrathin TiO_2_ nanosheets compensate for the volume fluctuations during intercalation to a great extent. In consequence, the anode showed higher rate capability (792.3 mA hg^−1^ at 100 mA g^−1^) and better cycle performance (805.3 mA hg^−1^ at 0.1 A g^−1^ after 100 cycles).

### 4.3. Three-Dimensional Structure

Three-dimensional nanostructures of TiO_2_ have drawn intensive attention from researchers in LIB research because of some advantageous characteristics like tunable pore sizes, interconnected structure, highly exposed surface area, and high porosity. These properties facilitate the electrochemical performance of the material as an anode. First, they provide short diffusion lengths for Li^+^ ions, which improves the diffusion kinetics. High surface area ensures more electrolyte infiltration into an electrode, resulting in more Li^+^ storage. Moreover, the void spaces introduced by the porous structures displayed the volume-changing effect during charging/discharging and maintained structural integrity. Furthermore, their interlinked structures show better electrochemical performance than loosely packed nanocrystals.

TiO_2_/oleylamine was used as a precursor to create porous TiO_2_ urchins with spherical cores and nanochain-constructed shells. The resulting TiO_2_ material has outstanding electrochemical performance, with high capacity, superior rate capability, and great cycle stability. Its increased electrochemical performance can be ascribed to its unique porous architectures and excellent structural stability. More crucially, ex situ structural characterizations show that trace anatase nanocrystals are irreversibly transformed into cubic Li_1_TiO_2_ nanodots, which explains the modest capacity degradation of the porous TiO_2_ material during long-term cycling. More work is underway to improve the geometry and pore structure of the TiO_2_ material for more robust cycle stability. For instance, porous TiO_2_ urchins and (protonated) titanates may have uses in sensitized solar cells, (photo)catalysis, and sorption [91]. A simple in situ hydrolysis technique is used by J.Y. Shin et al. [92] to successfully produce a hierarchical, nanoporous TiO_2_ structure. Because of its large surface area, it offers sustained high lithium-storage capability when used as an anode material, particularly at high charge/discharge rates. The material exhibits two distinct storage modes: (a) bulk insertion and (b) pseudo-capacitive interfacial storage responsible for 64% of the total capacity when cycling at high rates to emphasize the interfacial storage, yielding 302 mA hg^−1^ and 200 mA hg^−1^ of completely reversible discharge capacity kinetically at charge/discharge rates of 1C and 5C with extremely high cycle stability.

Specific surface area increases greatly when nanosynthesis is extended to the microsphere level. Fengyan Xie et al. [93] synthesized hollow TiO_2_ microspheres that comprised densely interconnected nanowires. These microspheres of 500–800 nm diameter possessed high specific surface area (131.92 m^2^ g^−1^). The urchin-like 3D microspheres have ~2.4, 1.5, and 2.8 times higher diffusion kinetics than the commercial TiO_2_ nano powder and the 0D and 1D TiO_2_ nanostructures because they comprised both 0D (nanoparticles) and 1D (nanowires) [94]. The loosely packed nanowires and nanoparticles centered by a core provide exposed surface area. Thus, the electrochemical performance subsequently improves.

Lui et al. [95] introduced a 3D ordered macroporous interconnected (3DOM) TiO_2_ structure through a simple in situ preparation technique where polystyrene was used as beads and carbon cloth as a template, and the material exhibited a high initial capacity of 402 mAh g^−1^, which reflected the contribution of carbon content in the high initial capacity. Moreover, voltage plateaus at this rate were observed from charge/discharge curves 1.79 V and 1.9 V with a less expanded plateau region and a high slope region. This implies better interfacial lithium storage attributed to the porous interconnected structure. At the high current rate of 73.5 C, the material still possessed 69 mA hg^−1^ because the porous structure maintained high structural stability.

Another unique homogeneous yolk-shell hierarchical TiO_2_ macroporous structure (HYSM) in Figure 13a,b was introduced by Wang et al. [96] through a simple solvothermal process. The prepared sample exhibited a high initial discharge capacity of 267.6 mA hg^−1^ at 0.25 C and maintained 191 mAhg^−1^ even after 40 cycles, showing strong electrochemical performance as anode materials for lithium-ion batteries. These outstanding electrochemical performances may be ascribed to the unique yolk-shell hierarchical structure’s short Li-ion and electron routes, as well as the ample elastic buffer area to tolerate the significant volume fluctuation.

Sacrificial templating is commonly used in synthesizing nanostructured TiO_2_. Among the methods is bio-ingredient templating, which has been investigated widely because living organisms are used as templates. It is noted that bio-templates contribute large specific surface area, a highly exposed skeleton, and high interfacial density. Bacteria, fungi, animal cells, plant cells, and other living prokaryotes and eukaryotes are used as templates that demonstrate feasibility as proteins, enzymes, and DNA as molecular subunits. Yeast and glucose have been individually reported as template contributors in the pore formation process of TiO_2_ nanostructures. Using yeast and glucose as bio-templates, a novel microporous anatase TiO_2_ (pore size: 2.5–3 mm) with mesopores (4–51 nm), sample PT, was effectively produced by Y. Chang [97] through a sol–gel method. The author collaborated to provide the final porous structure. The porous topology allowed for large surfaces and open spaces. Because of the easy intercalation/de-intercalation of Li, the material performed well as an electrode material for LIBs with high-capacity retention. A carbon-coated network significantly enhanced the electrochemical properties of porous TiO_2_. The hierarchical porous structure exhibited a high diffusion coefficient along with outstanding cycle stability and rate efficiency, which is required for rechargeable batteries to operate at high rates. 

A template-free hydrothermal process also resulted in a 3D hollow porous TiO_2_ microsphere. This microsphere as anode showed good reversible storage capacity (170 mA hg^−1^ at 0.6 C for 150 cycles) and rate capability (40.8 mA hg^−1^ at 24 C). The porous shells of the microspheres maintained closer contact with the electrolytes. Therefore, the diffusion lengths were shortened, the ion transfer mechanism was improved, and more lithium ions were accommodated in the porous shells. In addition, the aggregation of the active material was halted by the hollow architecture [98]. However, the electrochemical performance varies with the size of the hollow macropores even though the size of the microspheres is more significant than the nanoparticles’ sizes and their crystallinity. Xiaobing Wang [99] showed in his study that when microsphere size was increased from 0.62 μm to 1.20 μm, reversible capacity also increased from 81–94.8 after 100 cycles. Therefore, it is important to control microsphere size to obtain the required electrochemical performances. For instance, 3D hollow anatase TiO_2_ microspheres with highly exposed {001} energy facets also facilitate higher lithium storage capability by providing more active sites and can absorb the local volume-changing effects with the hollow infrastructure [100].

Doping is another strategy for increasing the electrochemical behavior of TiO_2_ as an anode for LIBs. Fe, Sn, Zn, and Nb have been investigated extensively. It has been proven in studies that N-doped TiO_2_ shows one of the most promising anodes in terms of performance and availability. Hasegawa et al. [101] have synthesized a three-dimensional N-doped TiO_2_ through Ti-based ethylenediamine macroporous organic–inorganic hybrid gels. The prepared N-doped TiO_2_ calcined at several temperatures (400 °C, 500 °C, 600 °C, 700 °C) under ambient conditions resulted in a hierarchal porous structure that comprised different-sized macro–micro pores with the interstices of TiO_2_ crystallites, whereas 500 °C calcined samples showed the best electrochemical performances. The samples displayed reversible discharge capacities of 252 mA hg^−1^ at 20 mA g^−1^ rate and 172 mA hg^−1^ at high 200 mA g^−1^ rate within 1.9–2.1 V. Praiseworthy cyclic performance of more than 200 mA hg^−1^ at 100 mA g^−1^ rate over 300 cycles was also observed. The improvement of the electrochemical behaviors can be attributed to both the porous hierarchal structure and nitrogen doping. The contributions of nitrogen doping are the improved electron conductivity as well as the improved Li^+^ ion diffusivity, although the amount is small. In addition, the nitrogen-distorted Ti-O lattice bond affects the electrochemical reactions at the interface of the electrolyte and the active material, which could influence performance. 

A simple and fast hydrolytic process has been reported by Di Lupo et al. [102] for synthesizing mesoporous TiO_2_ nanocrystal (TiO_2_-NC) where a tetrabutylammonium bromide surfactant was utilized. Its electrochemical behavior as a high-rate Li-ion battery anode at ambient temperature was effectively shown even after extremely long cycling (more than 1000 cycles). X-ray diffraction, transmission electron microscopy, nitrogen physisorption studies, and electrochemical methods were all used to fully describe the samples. A regular mesoporous TiO_2_ anatase phase with a specific surface area of 258 m^2^ g^−1^ and a high degree of crystallinity was synthesized using a new synthesis method without the need for further treatments. In addition, the material was calcined between 250 °C and 550 °C to enhance crystallization and test the impacts of structural changes on electrochemical properties. The produced materials’ potential for next-generation high-power lithium-based batteries is shown in their impressive rate capability and outstanding stability through extremely extended cycles.

## 5. Different Preparation Techniques

The aforementioned nanosizing greatly improves the electrochemical performance of TiO_2_. Jiang et al. [103] showed that 15 nm rutile nanoparticles possessed an initial discharge capacity about 378 mA hg^−1^ and a reversible capacity of 207 mA hg^−1^ after 20 charge–discharge cycles. Nanoparticles < 100 nm have shown their effects of surface tension on Li^+^ ion insertion thermodynamics [104]. The volume stress that occurs during intercalation is conveniently supported by the thermodynamics of nanosized particles rather than large [105]. A wide variety of techniques have also been noted for the nanosynthesis of TiO_2_ including solvothermal, hydrothermal, hydrolysis, electron spinning, and anodization. The morphologies of the TiO_2_ nanostructures with respect to particle sizes and internal architectures are mainly influenced by the preparation recipe followed.

### 5.1. Solvothermal

Liang et al. [106] synthesized TiO_2_ nanoparticles with average 10 ± 5 nm via one-pot solvothermal reaction. These nanoparticles as anodes for LIBs showed 196.4 mAhg^−1^ reversible capacity at 0.2C rate after 100 cycles, a clearly improved electrochemical performance compared with bulk. Following the same solvothermal technique, Jin et al. [107] produced hierarchical nanosheets that can sustain over 700 reversible cycles at 1 C rate as anodes for LIBs. Similarly, anatase/graphene nanosheets [108] and carbon-coated mesoporous TiO_2_@graphene nanosheets [109] were synthesized and showed better performance as LIBs anodes.

### 5.2. Hydrothermal

Gorparti et al. [110] synthesized carbon-doped TiO_2_ nanowires via a hydrothermal synthesis. TiC was used as the precursor, and the reaction was accomplished in NaOH solution. The resulting product as an active material for LIBs showed a higher reversible capacity of 306 mA hg^−1^ along with higher cycle stability (>1000 cycles) at 0.1C. The enhanced performances can be attributed to the presence of carbon, which not only decreased the diffusion length but also increased the electron conductivity. In addition, hydrothermally prepared TiO_2_@C hollow spheres [111] and TiO_2_ nanotube@graphene [112] showed 170 mA hg^−1^ and 357 mA hg^−1^ reversible capacities, respectively.

### 5.3. Hydrolysis

TiO_2_ nanoparticles@graphene prepared through hydrolysis exhibited higher cycle performance as it could withstand 400 cycles successfully at 2C rate and reversible capacity of 70 mA hg^−1^. Wang et al. [113] also synthesized anatase TiO_2_ nanowire via hydrolysis followed by 400 °C calcination. The nanowires showed promising electrochemical performance. After 40 cycles 280 mA hg^−1^, discharge capacity was available with 98% Coulombic efficiency. Gao et al. showed anatase nanorods displaying 206 mA hg^−1^ reversible discharge capacity at 5 Ah g^−1^ rate [114].

### 5.4. Electrospinning

Electrospinning is a popular method of nanofiber synthesis that has been applied widely. Tammawat et al. [71] used this method to produce TiO_2_ nanofibers. The nanofibers as anodes exhibited promising electrochemical performances. High initial discharge capacity of 325 mA hg^−1^, reversible capacity of 175 mAhg^−1^ at 0.3 C rate, and very low-capacity fading show its enhanced electrochemical efficiency. Hollow TiO_2_ and carbon composite nanofibers are also made by spinning technique. The synthesized products have 228.9 mA hg^−1^ reversible capacity after 100 cycles at 100 mA g^−1^, whereas it was only 61 mA hg^−1^ for non-hollow TiO_2_/C nanofibers [115]. Thirugunanam et al. [116] fabricated TiO_2_ nanofibers of 2–15 nm average diameter by a lost cost electrospinning method. The nanofibers are wrapped with two layers of reduced graphene oxide (rGO). The high rate capability of this synthesized nanofiber derives from the influence of Ti^3+^-C bonds, which anodically favored the electrochemical process.

### 5.5. Anodization

Anodization is a simple electrochemical method of nanofabrication that is widely used for TiO_2_ nanotube synthesis. Shanon et al. [117] prepared TiO_2_ nanotubes via anodization consisting of both anatase and rutile phase. High cycle performance is very noticeable here, where this anode can withstand about 1000 cycles at 0.1C rate and gave 140 mA hg^−1^ reversible discharge capacity. Moreover, the study revealed that the rate performance evolved rapidly in H_2_-endowed calcination. 3D freestanding TiO_2_ nanotubes were synthesized by Wei et al. [118] with a high aspect ratio. These nanotubes gave specific capacity of 96 mA hg^−1^ at 5C rate and were cycled successfully up to 500 cycles with only 6% capacity loss. Anodizing alloy Ti_80_Co_20_ self-organized nanotubes can be grown. The formed TiO_2_/CoO showed reversible areal capacity 600 μAhcm^−1^ at 10 μA cm^−2^. These formed nanotubes also possess double the lithium insertion capacity of formed nanotubes. High coulombic efficiency ~98% again confirmed the excellent electrochemical properties of these nanotubes as anodes [119].

In addition, there are other more methods used for TiO_2_ nanosynthesis that are later implemented as active materials in LIBs. The electrochemical performances of different forms of nanostructured TiO_2_ as anodes and their corresponding preparation methods are summarized in the following Table 1.

## 6. Nanostructured TiO_2_ by Electrochemical Anodization

Potentiostatic anodization is of great interest as a preparation technique for nanostructured TiO_2_. Nanostructures can be prepared by different methods pf electrochemical anodization as long as they consider the need for a highly ordered, closely packed structure with proper alignment. Controlled geometry was also achieved with self-templating method, and the cost-effectiveness and easy preparation have advanced anodization and attracted attention for wide applications [129]. Quan et al. synthesized a well-structured, closely dense TiO_2_ nanotube (TNT) of 30–90 nm pore diameter and of 300–500 nm length that was grown via anodization with optimized voltage (20 V) parameter and 0.2 wt.% HF aqueous electrolyte solution [130]. However, varying the anodization parameter to achieve different outputs has been reported. Anodizing pure Ti substrate with 4 wt.% HF electrolytes in organic solution, C. Ruan and his coworkers [131] obtained positive outcomes. A well-aligned TiO_2_ nanotube surface structure (60 nm inner pore diameter, 40 nm thickness and 2.3 μm length) formed in which a 20 V DC supply continued for 72 h. These results have led to research advances in the optimization of anodization parameters like electrolyte combination, temperature, agitation, applied voltage, anodization time, and pH. to realize the mechanism of nanostructure formation and to apply the nanostructures in research.

### 6.1. Anodization Process 

Basically, anodization of titanium is an electrochemical process where pure Ti substrate or titanium alloy acts as an anode, and collaterally, Pt or graphite acts as the cathode. Voltage applied for a certain time like in Figure 14 causes the formation of a bilateral oxidation layer and the dissolution of the oxide layer into electrolytes. When the reaction rate of these two are equal, a stable dimensional structure appears, resulting in different types of nanostructures.

Ti substrate exposed in water forms a compact layer of TiO_2_ within a few seconds because of the reaction between Ti and H_2_O molecules shown in Equation (3):Ti + 6H_2_O ⟶ TiO_2_ + 4H_3_O^+^ + 4e^−^(3)

The dissolution of this oxide layer starts parallelly because this layer is subjected to field-aided high polarization effect. As a result, Ti^4+^ cations are induced through the breakdown of Ti-O bonds. However, hydrogen evaluation is seen at the cathodic terminal shown in Equation (4):8H_3_O^+^ + 8e^−^ ⟶ 4H_2_ + 4H_2_O(4)

After applying the voltage, dissolution starts occurring at the anode, which induces Ti^4+^ ions according to Equation (5), and the deprotonated H_2_O splits into OH^−^ or O^2−^ ions that exist in electrolytes. These oxygen anions migrate from the oxide/electrolyte interface to the metal surface, whereas Ti^4+^ cations move from a metal/electrolyte interface to an oxide/electrolyte interface. Thus, an oxide layer forms according to Equation (6); this ion transfer is sponsored by an electric field:Ti ⟶ Ti^4+^(5)
Ti(OH)_4_ ⟶ TiO_2_ + 2H_2_O(6)

The presence of F^−^ species triggered a chemical attack on the oxide film subjected to pore initiation. First, a complex ion is formed between the interaction of Ti^4+^ ions and F^−^ species (Equations (7) and (8)):Ti^4+^ + 6F^−^ ⟶ [TiF_6_]^2−^(7)
TiO_2_ + 6HF^−^ ⟶ [TiF_6_]^2−^ + 2H_2_O + 2H^+^(8)

Finally, the forced chemical attack of the complex ions on the oxide layer causes pore initiation. The hole mechanism is portrayed in Figure 15a–e. 

The chemical attack of [TiF_6_]^2−^ causes small pits on the oxide layer like in Figure 15c. These pits are the results of a chemical-assisted electric field. As the anodization period proceeds, the pits become wider and turn into pores (Figure 15d) and then nanotubes (Figure 15e). It is noted that the formation of nanotubes is a competition between two reactions, oxide formation and the chemical etching of the oxide layer. A stable nanostructure layer is formed when the rates of these two reactions are equal. Higher anodization voltage is the cause of high oxidation rates and results in a thick layer of oxide [129].

### 6.2. Anodizing Parameters 

#### 6.2.1. Voltage

In anodization, manipulating the parameters of applied voltage, time, and electrolyte composition containing F^−^ allows for controlling the dimensions (wall thickness, length) of nanostructures. Mor et al. [132] introduced a tapered, conical TiO_2_ nanotube where the contribution of the voltage sweep rates appeared clearly on light. The effects of gradual increments of applied voltage from 10–23 V at 0.43–2.0 V/min rate are reported. At low 0.43 V/min voltage ramp, the lowest inner diameter of 65 nm was achieved, and it subsequentially increased with the increase in the voltage ramp. However, voltage applied in the reverse direction, from 23–10 V, at different sweep rates had no effect of a tapered shape. Moreover, nanotubes with uniform diameters appeared at constant voltage ramp. It is thus obvious that the applied voltage with increasing sweep rate influenced the gradual wider inward movements of the oxide layer and made a conical tubular structure.

Gong et al. [133] synthesized self-organized and self-growing TiO_2_ nanotube arrays of 25–65 nm tube diameter by anodizing pure Ti substrate where an array formed with high density. This study illustrates that as the voltage increased, the tube diameter increased commensurately, but after a certain limit of applied voltage, the nanotube structure became obscure. In addition, greater potentiostatic effort is needed for well-established tubular architecture in dilute HF electrolyte solution, which implies that the concentration of the HF solution has impacts on nanotube formation. 

Liu et al. [134] proposed a simple method of synthesizing a TiO_2_ nanotube following a two-step anodization method. In the first step with 60 V, 1 h anodization revealed a highly ordered nanotube array with tube diameter ~140 nm and wall thickness ~15 nm. The final step with 20 V 1 h anodization gave compact and small-diameter nanotubes without any noticeable defects. That slight variations in applied potentials can cause a change in tube diameter was also revealed by Bhardwaj et al. [135]. Here, the anodization voltage was varied: 50 V, 52 V, 55 V, and 57 V. The resulting tube diametera are 34 nm, 44 nm, 53 nm, and 58 nm, respectively. The tube length changes for 57 V only, showing the effect on tube length of higher voltage. Another study showed that as the voltage increases, both the length and diameter change commensurately: For 20 V, 25 V, 30 V, 35 V, 40 V, 45 V, and 50 V, the formed nanotubes diameters are 58.25 nm, 70.29 nm, 91.24 nm, 77.33 nm, 96.98 nm, 119.68 nm accordingly, and the corresponding tube lengths are 1.04 μm, 3.32 μm, 3.68 μm, 3.21 μm, 4.27 μm, and 8.35 μm, respectively [136]. Optimum voltage tuned with the proper time causes improved geometry regarding both length and diameter. In addition, 60 V 3 h anodization results in high 46 μm thick nanotubular structures with 120 nm tube diameter [137], whereas, 30 V 45 min anodization results in 1.2 μm thick nanotubes [138]. High applied voltage causes spark anodization on the Ti surface [139]. S. Park et al. [139] showed that after 230 V anodization, the sparks that occurred resulted in a di-electric breakdown. The pore size dramatically increased to 1–2 μm due to the dissolution of the oxide layer, but breakdown voltage specifically depends on the electrolyte composition. Applied voltage has a direct correlation with increasing dimensions (pore size and length), as revealed by M. Kulkarni et al. [140], where the trend was observed for increments of potential from 10 V to 70 V.

#### 6.2.2. Electrolyte Composition

Zwilling et al. [141] reported on pure Ti substrate and alloy TiA6V (Ti-6%, Al-4% v) through anodic oxidation in the presence of chromic acid (CA) of 0.5 mol/L with and without HF of 0.0095 mol/L as a source of F^−^ ions. It is noted that F^−^ ions contribute to the growth of porous structures as no porous layer was observed without the presence of HF in CA electrolyte solution. F^−^ introduces pore initiation and consequently etches away metal ions from a pore bottom of metal or alloy substrate where the breakdown of the thin compact layer that formed in the CA solution at 3 V/s is repaired by an associated high current. Both the compact layer and the duplex porous layer thicken from the residual current. The overall electrochemical efficiency is not more than 50%, and it drastically falls with the increase of the anodizing period. The Cr species of CA solution played an adverse poisoning role, whereas F^−^ species worked as antidot in growth dynamics. 

Electrolytes play an inevitable role in anodization mechanisms; modifications of electrolyte solution are observed over days to find required feedbacks. It is noted that highly ordered titanium nanostructures can be effectively grown at room temperature and are possible through anodization only by ensuring the source of F^−^ in the electrolyte [142]. A short nanotube texture appeared in pure 0.5 wt.% HF electrolyte solution. A neutral electrolyte solution of Na_2_SO_4_/(NH_4_)_2_SO_4_ and NH_4_F caused longer tubes up to 2.5 μm as the dissolution rate of Ti^4+^ ions from Ti substrate is higher in this type of solution [134]. F^−^ based organic electrolyte containing some organic solution such as ethylene glycol (EG) overcome the roughness issue. However, the presence of EG slows the ions’ movements, and thus, the etching of metal ions happens in a gentle smooth manner. EG is a highly viscous fluid with a high degree of homogeneity that lowers the diffusion constant of electrolytes. By confining the pH burst to a specific location and suppressing the concentration in a localized manner, EG initiates pits. This process results in smoother, much longer tubes up to 1000 μm with a high aspect ratio, whereas this organic solution implanted for multistep anodization process converted the oxide film into a hexagonal tubular architecture. In addition, using F^−^-free electrolyte such as HClO_4_ resulted in disordered and shorter nanotubes [143]. Optimizing the amount of F- species is essential. Electrolyte containing extremely low wt.% of F^−^ species (<0.5 wt.%) resulted in a non-homogeneous porous structure where the pore distribution was also scattered and unevenly distributed over the Ti substrate. The best result appeared for 0.5–1 wt.% with a highly ordered and thick (2.5 μm) porous structure, while more than 1 wt.% exhibited no considerably improved structure as the solubility of F^−^ ions acted as a limiting factor. Rojas et al. [144] reported a comparison between electrolytes of Na_2_SO_4_ and (NH_4_)_2_SO_4_. Na_2_SO_4_ electrolytes containing 0.5, 1, 3, and 5 wt.% NH_4_ produced nanotube structures with more distinct dimensions than the structures produced in (NH_4_)_2_SO_4_ electrolyte containing 0.5, 1, 3, and 5 wt.% NH_4_. Moreover, pure nanoporous structures were observed for 0.5 and 1 wt.% of NH_4_ in both electrolyte solutions. Rahman et al. [145] produced different nanoporous structures in electrolyte solution containing 0.5 wt.% NH_4_F and 1 M (NH_4_)_2_SO_4_. Here, only variations in ethylene glycol (EG) content—5 vol%, 10 vol%, 30 vol% and 50 vol%—are the main precursors for forming different sizes of nanopores. Dimethyl sulfoxide (DMSO) electrolyte with 1 wt.% HF, 2 wt.% HF, 4 wt.% HF, and 6 wt.% HF produced nanotubes of 53 μm, 13 μm, 53 μm, and 52 μm tube length, respectively, where no obvious effect on pore diameter is visible. Here, the prolonged 70 h anodization period resulted in contained fluoride species. As a result, the diameter converged to saturation [146].

Electrolytes not containing fluoride ions have also been used to synthesize TNTs [147]. Anodization in perchlorate and chloride-containing electrolytes used by Hahn et al. [148] produced TNTs with a superior aspect ratio where the electrolytes contained oxalic, formic, or sulfuric acid electrolytes with chlorine ions. Chlorine-based (fluoride-free) electrolytes provide an advantage over those that include fluoride. When using a fluoride-free electrolyte, long TNTs may develop in a fraction of the time it takes in the fluorine-based medium (for example, 17 h to 10 min), rendering the rate of TNT formation a thousand times faster. A bromide-based solvent was also found to evolve rapidly in TNTs, but self-organized and shooting alignment was obscure [149]. Reutilizing electrolytes that have been employed for prior anodization also exhibited positive results with better nanostructures [150]. Electrolytes used for a second time showed improved pore diameter whereas wall thickness decreased, which resulted in converging the higher chemical etching rate that, which was attributable to the increase in electrolyte conductivity when it is used for further anodization. Consequently, it accelerates the reaction rates and plays a role in better TNT formation [151].

#### 6.2.3. Electrolyte pH

It is worthy to mention that pH is one of the most important factors in anodization, and manipulating the pH value can control the chemical dissolution of TiO_2_. Higher pH increases chemical etching and bilaterally lessens the oxide layer thickness. An acidic environment accelerates the corrosive properties of metal, resulting in higher metal loss. Several factors influence the electrolyte’s pH during anodization [152]. The local acidity is caused by hydrolysis production at the working electrode. Longer tubes were made possible by tailoring the local acidity, which encourages TiO_2_ breakdown and provides a more protective environment against dissolution near the tube opening. The counter electrode is where hydrogen formation and the production of OH^−^ species take place [153,154]. Kang et al. synthesized nanotubular structures with different tube lengths using electrolyte solution 0.5 wt.% NaF + 0.5 M Na_2_SO_4_ + 0.5 M H_3_PO_4_ + 0.2 M Na_3_C_6_H_5_O_7_ + NaOH at pH of 1, 3, 4.2, and 5 [155]. Joseph et al. [156] have produced TiO_2_ nanotubes of different dimensions by varying the pH when the other parameters were kept the same. Pure Ti substrate was anodized in ethylene glycol solution containing NH_4_F (0.35 g) and DI water. Current of 30 V was applied for 5 h, and pH ranged from 2 to 4. Finally, the anodized nanotubes appeared with tube lengths 3.5 μm to 5.2 μm. When pH was increased to 10, the tube length also increased to 5.6 μm, but at pH 12, the length decreased to 1.8 μm. Again, the anodization of Ti foil in a NH_4_(SO_4_)_2_ solution containing 0.15 M NH_4_F created nanotubes where the tube length was as varied as 198 nm, 267 nm, and 295 nm at corresponding pH of 2, 3, and 4 [157]. Hence, it can be concluded that after a certain increment of pH, the tubular structures start to collapse because the H^+^ ions become contained at high pH.

#### 6.2.4. Current Density

The metal oxide dissolution rate and the pore size are affected by the various current densities. The electrochemical etching rate, power, and electric field intensity are all directly commensurate with the current density. Pits appear to expand prior to the development of channels, which divide the pores. The pore size of TNTs rises together with the increase in Ti foil current density. As a result, changing the current density might lead to various tube widths [158]. Aqueous electrolyte makes a simple pathway for current flow. This is why current density is higher in this case than in organic solution [159]. Optimized applied voltage and time scale combined have positive effects on anodization outcomes. Applied potential of less than 50 V turns the porous oxide into a self-organized and closely packed texture. Short anodization time results in an ordered array of tubular oxide film of a few hundred nanometers in length and tens of nanometers in diameter. Ripples formed between the connective junctions of these nanotubes, but as the time elapsed, these ripples disappeared as the TiO_2_ dissolved. In addition, applying voltage equal to or more than 50 V converts the layer in localized burst morphology to resemble coral reefs, as shown in Figure 16a,b. Separation of the ordered array structure occurred at these higher potentials [160]. Bio-sensitive TiO_2_ hexagonal nanotubes are grown on Ti6Al4V and Ti6Al7Nb alloys in both organic ethylene glycol and glycerol solution. However, the annealed Ti6Al4V sample exhibited a homogenous and uniformly distributed oxide layer that comprised both rutile and anatase phase at 550 °C. The sample processed in glycerol possessed more bio-sensitive properties [161].

#### 6.2.5. Electrolyte Temperature

Electrolyte temperature is also an influence on the nanostructures of anodic TiO_2_. Nanotube length along with wall thickness decreased with the increase in electrolyte bath temperature. Similar cases were observed in a research study by Li et al. [162], where the average nanotube length decreased from 430 nm to 230 nm and the wall thickness from 21 nm to 15 nm when the electrolyte temperature increased from 10 °C to 35 °C. On the contrary, pore diameter increased from 58 nm to 81 nm, which conflicted with the findings by Mor et al. [163] of no obvious relationship between pore diameter and the other anodizing parameters except applied voltage. Moreover, after a certain temperature, the nanostructure can change its pattern because high temperature eventually increases the velocity of F^−^ species attack in scattered directions. As a result, high disoriented chemical dissolution returns in noodle-like nanostructures. Furthermore, the number of pores subsequently increased with electrolyte temperature due to lessening the viscous effects and accelerating chemical attack [164,165].

#### 6.2.6. Time

Anodization time plays a significant role in nanostructure formation. No nanostructured is formed in a too-short anodizing period [166]. At least 15 min is required to form nanotube, and at least 5 min is required to form nanopores. In addition, short nanotubes can be acquire in a 5 min anodizing period [167]. However, 7 μnm long nanotubes were also formed in 25 s in the presence of lactic acid-based electrolytes. This lactic acid strongly prevents the di-electric breakdown and allows for faster ion transports, which in turns results in high growth [168]. Park et al. [139] showed that anodization in the presence of EG containing 0.3 wt.% NH_4_F and 2 vol% H_2_O results in highly ordered nanotube structures, where for 20 min, 30 min, and 40 min anodization periods, the corresponding tube thicknesses were 18 μm, 25.2 μm, and 30.8 μm. A proportional relationship is established between time period and nanotube thickness. It is noted that after an extended period of anodization, no significant impacts on tube thickness or depth are seen [133]. A prolonged period of 19 h causes nanotubes of 1.3 μm thickness, again supporting this argument [169].

#### 6.2.7. Counter Electrode

A counter electrode (CE) is an important parameter that influences the morphologies of the oxide layers. Anodization was investigated with different CEs like iron, stainless steel, carbon and aluminum by Sreekantan et al. [170]. Short nanotubes with nonuniform wall thickness (5–10 nm) were found for stainless steel CE. The aspect ratio obtained with an aluminum cathode is the same as the aspect ratio obtained with a carbon cathode. The top surface of the TNT produced by the aluminum cathode, on the other hand, is comparable with that produced by the stainless-steel cathode. The walls at the bottom of these TNTs appear to be sturdy, but the top is prone to collapse. TNTs produced with an aluminum cathode are far less stable than TNTs produced with a carbon cathode. When an iron cathode is utilized, a well-organized TNT with high aspect ratios is generated. They are, however, less robust than those made with a carbon cathode. A carbon cathode results in nanotubes with greater aspect ratios, larger tube diameters, and longer tubes than other cathode materials. In addition, these nanotubes are excellent compared with the nanotubes formed using platinum cathode and are also less expensive. Allam et al. [171] synthesized TNTs using a wide range of CEs including Ni, Pd, Pt, Fe, Co, Cu, Ta, W, C, and Sn. Their findings indicated that the composition of the cathode material has a significant impact on the appearance of the surface precipitate. The cathodes’ overpotential is a significant component that affects the dissolution kinetics of the Ti anode, which in turn controls the activity of the electrolyte and the morphology of the produced TNT. The more Ti that is dissolved in the electrolyte, the greater the conductivity of the electrolyte, which helps to reduce debris development. Due to variances in their overvoltage inside the test electrolyte, it appears that the diverse cathode materials resulted in the creation of distinct morphologies. Consequently, the cathode materials are arranged in the order Pt = Pd> C > Ta > Al > Sn > Cu > Co > Fe > Ni > W based on their sustainability in aqueous electrolytes.

The distance between two electrodes is also a great influencing factor in anodization, as revealed by Yoriya et al. [172]. Closer inter-electrode distance improves electrolyte conductivity and titanium concentration. Therefore, the pore dimensions improve subsequently. Optimum distances of 0.5 cm to 4 cm are maintained to achieve well-developed, highly ordered nanoporous structures. Different nanotubes by anodic oxidation are summarized in Table 2, which briefly presents the variations in the dimensions with the varying anodizing parameters.

## 7. Anodized TiO_2_ as a Promising Anode for LIBs

The electrochemical anodization of Ti results in differently nanostructured TiO_2_, especially nanotubes, which offers the option of altering the dimensions of the structure by altering the anodizing parameters. A well-aligned, self-organized TiO_2_ nanostructure array widens the possibility of being implemented as a potential anode for LIBs. These nanostructures, having high surface area, open the door to desired electrochemical properties by decreasing the Li^+^ ions pathway, increasing ionic conductivity, and offering structural stability during Li^+^ insertion/deinsertion. Then, the abatement of the polarization effect at the anode happens, and this triggers the high charge discharge capacity [129].

### 7.1. Morphological Impacts of Nanotubes on Electrochemical Performances

The electrochemical performances are directly related to the morphologies of the nanostructures [174]. The maximum bulk capacity of the Li^+^ insertion coefficient is 0.55 (Li_0.55_TiO_2_) [175], with the highest for nanosized TiO_2_ under 10 nm [176]. Among the different anodically fabricated nanostructures, nanotubes are the most common. The geometry of nanotubes reflects the electrochemical properties of LIBs. Specific capacities of anodized anatase TiO_2_ nanotubes are mainly the combination of their pseudocapacitive behaviors and bulk intercalation capacities, which derive from the architecture of nanotubes [177,178]. However, Zhu et al. [179] reported no direct relationship between nanotube lengths and rate performances, many studies revealed the relationship between the morphologies of nanotubes and their electrochemical performances [180]. Volumetric energy density is decreased with increasing nanotube diameter [171]. Specific capacity can be improved by increasing nanotube length and reducing diameter and thin wall thickness [181,182,183,184,185,186,187]. It is noted that long tubes ensure higher electrode electrolyte interface, which in consequence increases the areal specific capacities, but the higher rate performances are observed with shorter nanotubes. Because the longer nanotubes possess unstable architecture, the lengths are covered with nanograss. Moreover, the thicknesses throughout the lengths are not constant. As a result, the collapse of the tube lengths happens in some unstable points. The top of the nanotubes shows more chemical etching due to the higher anodization period. As a result, the openings are collapsed and appear like a bundle of nanograsses instead of tubes [182,183]. These nanograsses significantly affect lithium insertion capacities by varying the surface charge and the electrochemical double layer issues. In addition, the nanotubes formed in the organic electrolyte solutions appeared with an inner carbon rich layer, which also significantly decreases the lithium insertion efficiency. The composition of the carbon-rich layer is characterized by grazing incident scattering technology [184,185]. In order to overcome the effect of the carbon-rich layer, one effective approach can be applied: The inner carbon double layer can be dissolved using aqueous Na_2_SO_4_, which in consequence causes cathodic polarization [186].

The mutual dependency between specific surface area and discharge capacities are well established (Figure 17). Moreover, nanostructures possess higher specific surface areas than bulk. Again, the dependency of specific surface area on calcination temperature and crystal structure is strongly proven. A 300 °C calcined nanotube possessed a higher, 78 m^2^ g^−1^, specific surface area than the other amorphous calcined at 400 °C (70 m^2^ g^−1^) and 500 °C (57 m^2^ g^−1^). In addition, higher discharge capacities at different cycles for the 300 °C calcined nanotubes than other products are revealed to be associated with some crystal defects (Figure 17). Anatase is the most conductive TiO_2_ nanotube layers and can be achieved by annealing at 450 °C to 550 °C; in contrast, products annealed at more than 550 °C cause significant decreases in conductivity because the rutile phase is less electrochemically efficient. Furthermore, prolonged heat treatment at these favorable temperatures is also responsible for more conductivity [187].

### 7.2. Impact of Nanotubes Exposed Energy Facets on Electrochemical Performances

Highly exposed [001] nanotubes showed higher lithium insertion capacities than the nanotubes of arbitrary orientation because the exposed [001] orientations provide simple pathways for lithium insertion in the c-axis in anatase TiO_2_ nanotubes [188,189]. The anodization of polished Ti foils in aqueous electrolyte solution containing 2 wt.% ethylene glycol results in a c-oriented crystalline nanotube structure, where the unpolished Ti foils produce randomly oriented crystalline nanotubes with other residuals of the water-based electrolytes [190,191]. In addition, highly c-exposed nanotubes are formed when amorphous nanotubes are calcined in an inert medium [187]. The annealed products also have high oxygen vacancy (V-O) concentrations along with high preferential crystal sites. These both contribute to the higher rate capacities and enhanced lithium insertion/extraction capacities [192,193,194]. Auer et al. [195] also showed that completely V-O-free nanotubes can be produced along with preferential crystallographic sites [001]. The preparation technique includes the anodization of pure Ti foils where 2 wt.% water-based electrolyte was used followed by 450 °C calcination. As a reference, randomly oriented nanotubes were prepared in 10 wt.% water-based electrolyte that possessed same morphologies and almost identical thermodynamic properties. The reports say that the preferentially oriented nanotube crystals have higher storage capacities and rate capabilities than randomly oriented nanotubes without having oxygen vacancies. The fabrication of TiO_2_ nanotubes was extensively investigated by Kashani et al. [196] testing different counter electrodes; Pt, graphite, Ni, Ti, Cu, and Al. The synthesized nanotubes were heat treated at 500 °C for 3 h. As a result, all the samples for all counter-electrodes turned into anatase phases in the XRD pattern. The peak intensity ratios were 2.1, 1.7, 1.5, 1.0, 2.5, and 0.9 for graphite, Ni, Ti, Cu, Pt and Al, respectively. Hence, it is revealed from the study that the Pt electrode caused a highly exposed [001] anatase phase. Some rare defects in the nanotube structures were evidenced that can be attributed to the escaping of the water content in the heating process (250 °C to 500 °C) [169,197,198]. The nanotubes showed better rate capabilities, good cyclabilities, and high reversible capacities, and the structure also provides easier channels for Li^+^ insertion/deinsertion.

### 7.3. Doping

As a semiconductor material, TiO_2_ possesses a wide band gap (rutile ~2.98 eV, anatase ~3.05 eV, and brookite ~3.26 eV) under neutral conditions [199]. As a consequence, low electron conductivity limits it wide applicability in, for instance, high-performance lithium-ion batteries [174,191,199,200]. The inferiority in both ion and electron conductivity results in low-rate performances of LIBs [201,202,203]. Some possible solutions such as doping emerged to overcome this issue. Doping the nanotubes with C [204], Ni^2+^ [205], V^3+^ [206], Sn^4+^ [201], N [207], or Ti^3+^ [203] surprisingly improves ion electron conductivity.

#### 7.3.1. Self-Doping by Annealing

Annealing at high temperature is a convenient way to introduce reduced TiO_2_ nanotubes. Proper annealing atmosphere here substantially influences the conductive behaviors of TiO_2_ nanotube arrays with small alterations in morphology, size, valance band structure, etc. Volume expansion is buffered if the annealing conditions include a reductive ambience containing 5% H_2_ and 95% Ar and in consequence are subjected to simple lithium-ion intercalation/deintercalation. A sufficient number of oxygen bubbles absorbs the Li-Li repulsive effects, distorting the c-length of the unit cells to sufficiently improve high-rate capability [206]. The combination of 16% CH_4_ and 20% H_2_ in an annealing environment results in a carbon-modified TiO_2_ nanostructure as reported by Mole et al. [207]. This double-layer nanostructure showed a considerable increase in charge capacitance at optimized 40 V. Annealing with Ar only Ar showed better electrochemical properties than air with a high initial discharge capacity of 227.9 mA hg^−1^, significant capacity retention, and improved rate capability and cycle stability; the 450 °C annealed anatase structure was proven to be the most useful as anatase is the most advantageous crystal structure among all others for TiO_2_. In addition, the presence of a small amount of carbon, 5.22%, is more beneficial than the annealed air. An increase in the annealing temperature up to 600 °C or 800 °C did not improve the electrochemical properties as the anatase crystal disappeared [44,208,209]. Annealing a carbon-coated anodic nanotube structure (TiO_2−x_-NTs) in Ar and Ar/C_2_H_2_ medium was reported [191]. This structure was also oxygen deficient (TiO_2−x_-CNTs) and showed improved specific capacity of 320 ± 68 mA hg^−1^ (Li_0.96_TiO_2_); it was 180 ± 38 mA hg^−1^ (Li_0.54_TiO_2_) for TiO_2−x_, NTs. Furthermore, TiO_2−x_-CNTs have greater rate capability than TiO_2−x_ NTs, indicating that TiO_2−x_-CNTs are attractive anodes for high-energy and high-power LIBs. Cyclic voltammetry observations, as we all know, may be used to assess double-layer capacity. TiO_2−x_ and TiO_2−x_-CNT have double-layer capacitances of 85 μF cm^−2^ and 20 μF cm^−2^, respectively. The charge transfer resistance of TiO_2−x_-CNT is negligible at the electrode/electrolyte interface, as shown under electrochemical impedance spectroscopy, facilitating the movement of Li^+^ from the electrolyte to the electrode. In fact, cationic vacancies can be formed during TiO_2_ annealing in addition to oxygen vacancies. To investigate the production and impacts of point defects in TiO_2_ (oxygen and cation vacancies) caused by annealing anatase TiO_2_-NT in various atmospheres, O_2_, Ar, and N_2_ have been studied extensively. 

Annealing in an oxygen-poor atmosphere introduces more vacancies of oxygen, and free electrons can be formulated by the following Equations (9) and (10)
(9)$$TiO_{2}\;⟶\;2V_{Ö}^{\prime \prime }+O_{2}\;{}_{\left(g\right)}+{\mathrm{Ti}}_{\mathrm{Ti}}^{x}+4e^{-}$$
(10)$$TiO_{2}\;⟶\;2V_{Ö}^{\prime \prime }+O_{2}\;{}_{\left(g\right)}+{\mathrm{Ti}}_{\mathrm{Ti}}^{\prime }+3e^{-}$$
where $V_{Ö}^{\prime \prime }$ is the oxygen vacancy, and ${\mathrm{Ti}}_{\mathrm{Ti}}^{x}$ and ${\mathrm{Ti}}_{\mathrm{Ti}}^{\prime }$ are the reducing Ti. These oxygen vacancies are the main reasons for the higher electrical conductivity, larger capacity, and better rate performance [210,211,212,213]. Sava et al. [214] reported that oxygen vacancies improve the electron conductivity, whereas it is decreased slightly by higher Ti vacancies. These cation and anion vacancies are calculated using density functional theory. Raman spectroscopy also reveals that oxygen and Ti vacancies improve the areal specific capacities. Ar and N_2_ treated exhibited 10% and 25% improved capacities, respectively, and that for WV treated was 24%. These phenomena can be attributed to the fact of the increasing oxygen vacancies. In addition, the improvement in the electrochemical properties can be demonstrated by the improvement in the Li^+^ ion diffusion kinetics. This can be presented with respect to Ti vacancies. In short, the enhanced electrochemical properties are a combination of improved ionic and electronic conductivity [213].

#### 7.3.2. Electrochemical Self-Doping

Apart from annealing, another method of doping was recently introduced, electrochemical doping, where the reduction of TiO_2_ nanotubes is performed electrochemically [203,215,216]. In a two- or three-electrode anodization system, TiO_2_ nanotubes were used as the cathode, and Pt was used as the anode/counter electrode. The reduced TiO_2_ nanotubes appeared with higher lithium storage capacity and photocatalytic efficiency because the field-assisted potential amends the electronic structure of TiO_2_ and improves the electrochemical efficiency [217,218,219,220]. It is noted that the effect disappears very shortly after withdrawing the applied biased voltage due to the low viscous aqueous electrolyte used during reduction [218,219,221]. The doping of anatase TiO_2_ in an ethylene glycol electrolyte with protons was reported by Li et al. [222]. This black self-doping of TiO_2_ displayed its beneficial reduction effects over one year by reducing the band gap and consequently improving the electrochemical performances.

Some potential applications of nanotube TiO_2_ reduced via cathodic doping at elevated room temperature have been implemented successfully in field emissions [221], photocatalysis [214,220], and supercapacitors [214,220,222], but their applications are merely observable. Duan et al. [203] fabricated a Ti^3+^-doped 3D ordered TiO_2_ nanotubes that were later implemented as anodes for LIBs, and the anodes showed the expected improved electrochemical performances. This improvement is ascribed to the modified electronic structures of TiO_2_ nanotubes. The modification comes from the field-driven interactions of H^+^ ions, which in turns produce oxygen vacancies (V-O), OH groups, and Ti^3+^ ions. As stated previously, they contribute to the enhanced electron conductivity, lithium-ion diffusion kinetics, and increased storage capacity.

Bubble generation is a serious issue in cathodic doping. The large bubbles rupture and collapse the nanotube structures. In addition, the bubble formation causes voltage increment. The high voltage can demolish the structure. In addition, they force the H^+^ ions to modify the electronic surface of nanotubes. As a result, oxygen vacancies are created, and Ti is reduced to Ti^3+^ [214]. Hence, a great challenge must be faced in controlling the bubble formation, high voltage generation, and the enhancement of the electrochemical nature of TiO_2_ nanotubes. Another factor in accelerated bubble formation is low-viscosity electrolytes. A suitable ion kinetics into nanotube structures can only possible by ensuring suitable electrolyte viscosity [219]. A pulse voltage can produce in ethylene glycol electrolyte doping, subsequently improving the electrochemical performance [213]. The 4 min pulse voltage in turns exhibited well-structured tubular black nanotubes. This feature appeared with V-O/Ti^3+^ and OH groups. The doped black TiO_2_ nanotubes exhibited four times higher initial discharge capacities than the undoped. Moreover, the doped tubes showed high electronic conductivity and diffusion rates compared with the undoped. 

The surface characteristics and chemical environment of the prepared undoped and cathode-doped titanium dioxide nanotubes were investigated using X-ray photoelectron spectroscopy (XPS), electron paramagnetic resonance (EPR), and X-ray absorption near-edge spectroscopy (XANES) to reveal the reasons for the excellent electrochemical performance of self-doped nanotubes (Figure 18). According to Figure 18a,b, the doped TiO_2_ NTs showed greater concentrations of Ti^3+^/(V-O) and OH than undoped TiO_2_. Furthermore, the transfer of H^+^ species from the counter-electrode to the TiO_2_ NT electrode during the cathodic pulse doping process causes these additional defects in self-doped NTs. These H^+^ species convert some Ti^4+^ to Ti^3+^, leaving some oxygen vacancies and forming additional OH in the self-doped Ti^4+^.

The reduction reaction of TiO_2_ nanotubes can be presented as Equation (11): Ti^4+^O_2_ + H^+^ + e^−^ ⟵⟶ Ti^3+^O(O–H)(11)
where TiO_2_ is reduced to trivalent Ti^3+^ and created oxygen vacancies. At the same time, the Ti^3+^ becomes the center of the electron donors. With the increase in Ti^3+^ donor density, the electrochemical performances are increased [220]. When H^+^ and Li^+^ ions are in the appropriate electrolyte, reversible intercalation into and from TiO_2_ NT anodes occur [212,213]. As a result, the Ti_3_^+^O(O–H) groups produced in Equation (11) can store Li^+^ as well, and the appropriate mechanism is represented as follows in Equation (12):Ti^3+^O(O–H) + Li^+^ + e^−^ ⟵⟶ Ti^3+^O(Li–O) + H^+^ + e^−^(12)

In addition, the EPR and XANES spectra from Figure 18c–e show that the self-doped TiO_2_ nanotubes have more OH and O vacancies, which support the findings regarding the earlier XPS spectra. In short, the dense OH groups, V-O, and Ti^3+^ are the main reasons for the remarkable improvements in the electrochemical performances [223,224].

#### 7.3.3. Doping by Foreign Materials

Different foreign materials have also been used conveniently for doping TiO_2_ nanotubes. Positive outcomes regarding electrochemical performances can be observed here. Carbon is a very common foreign species for doping. It is a very good conductive reagent, and it enhances the overall electrochemical performances to a greater extent in carbon-doped TiO_2_ nanotubes (C-TiO_2_ NTs). Calcined TiO_2_ NTs in a mixture of C_2_H_2_/N_2_ atmosphere at 500 °C displayed higher cycling performance, superior capacity, and better rate performance than undoped TiO_2_ nanotubes [44]. 

Doping with N_2_ has been another fruitful way to improve electrochemical performance [225,226]. Zhang et al. [227] prepared N_2_-doped TiO_2_ nanotubes (N_2_-TiO_2_ NTs) via annealing TiO_2_ nanotubes under high NH_4_ gas flow. The synthesized product also showed better electrochemical performance such as marked improvements in rate performance. When cycling was performed from 0.1C to 10C, the capacity degraded only 200 mA hg^−1^ to 100 mA hg^−1^. However, the TEM images revealed that the chemical evaluation and structure were almost identical for both doped and undoped and the presence of N_2_ influences the simple lithium insertion into the nanotube structure. This contributes greatly to the improvement of both ion and electron conductivity and leads to higher electrochemical performance [228].

Sn-doped TiO_2_ nanotubes can be formed by anodization of co-sputtered Ti-Sn alloy in an electrolyte solution containing 1.3 wt.% NH_4_F and 10 wt. % H_2_O [229]. The formed nanotubes are a Ti_1−x_Sn_x_O_2_ type where the Ti^4+^ are substituted with Sn^4+^ [230] Here, the presence of Sn^4+^ caused an anatase-to-rutile phase transformation at the beginning when the lithium insertion coefficient x = 0.05 and full transformation occurred when x became 0.5 [210,211]. The rutile phase allows for the high diffusion of lithium and improves the electrochemical properties [199].

Nb doping has recently gained considerable attention. Nb-doped mesoporous TiO_2_ nanotubes were fabricated by Wang et al. [231]. The highly concentrated Nb-doped TiO_2_ nanostructure was synthesized by anodizing Nb/Ti alloy. Moreover, the Nb^5+^ replaces the Ti^4+^ cations and endows the upper surface of the nanotubes. Nanotubes containing 10 wt.% Nb concentration showed about twice the discharge capacities of the undoped. Furthermore, the higher cyclic performance at multiple high C rates and 88% capacity retention even after 100 cycles proves the contribution of Nb to the enhanced electrochemical performance of Nb-doped TiO_2_ nanotubes [232].

#### 7.3.4. Amorphous and Anatase TiO_2_

Anodically prepared amorphous and crystalline TiO_2_ nanotubes were investigated by S. Ivanov and his coworkers [233]. Electrochemical investigations were carried out in two different electrolyte media: 1 M LiPF6 ethylene carbonate/dimethyl carbonate (EC: DMC) and in 1-buthyl-1-methyl pyrrolidinium, (trifluoromethyl) sulfonylimide ([BMP][TFSI]) containing 1 M Li [TFSI]. Slightly higher electrochemical efficiency was observed for the latter case due to its inertness toward the anode surface and high stability. However, for the first type, a side reaction with water causes instability. Both amorphous and crystalline showed high charge/discharge capacities of 180–200 mA hg^−1^ for the open hierarchal structure. Amorphous samples exhibited 100% capacity retention, while capacity fade was observed in the crystalline structure. This can be attributed to the fact of defects detected in the surface due to heat treatment. Again, highly stable nanotubes were synthesized using ionic liquid (IL) during anodization instead of conventional NH_4_F, EG (AF) based electrolytes by H. Li et al. [234]. The microstructural cracks on the calcined anatase crystal TiO_2_ nanotubes shown in Figure 19a,b are associated with carbonaceous products. This fact can be attributed to the rapid decomposition of the unstable AF electrolytes that form a composite with the anode. IL nanotube products showed excellent electrochemical behavior, including capacity retention over more than 1200 long cycles; the sample remained intact with mild material loss due to its highly conductive nature, which resulted in a slow decomposition rate. Meanwhile, for the AF-based products, capacity deteriorated after 380 cycles, indicating structural disintegration.

The adverse effects of carbon contamination in nanotube structures are considerably pronounced. Kirchgeorg et al. [235] have synthesized a self-organized, highly ordered nanotube array with 32 μm tube length that exhibited less insertion efficiency. The efficiency decrease can be attributed to the longer tubes as the greater lengths contain more carbon contamination. The double-layer effect combined with a nanograss covering is responsible for the surface charges. The dissolution of carbon with ethylene glycol solution mixed with dimethyl sulfoxide and aqueous Na_2_SO_4_ increased the charges 33% and 17%, respectively.

Ryu et al. [176] presented a comparative analysis between anodically prepared amorphous and anatase TiO_2_ nanotubes, and no distinct voltage plateaus were observed for amorphous. Meanwhile, clear voltage plateaus were visible for anatase at 1.73 V and 1.9 V. However, the initial discharge capacities were 0.207 mA hcm^−2^ and 0.110 mA hcm^−2^ for amorphous and anatase, respectively, showing the superiority of amorphous in this regard. This means that the amorphous nanotubes contain more lithium storage capacity per mol TiO_2_ than anatase. This obvious supremacy can be attributed to the partially reversible surface lithiation [236] and the better pseudo-capacitive behavior of amorphous. Moreover, the better cycle performance of amorphous than that of anatase may be due to the improvement in structural stability during cycling [237]. The irreversible capacities were about 5% and 35% for anatase and amorphous, respectively, and the high reversible capacity loss for amorphous can be attributed to the absorption of H_2_O and OH by the irreversible sites of amorphous. In addition, it can be noted that the amounts of absorbed H_2_O and OH by amorphous TiO_2_ are 8.3 wt.% and 6.4 wt.%, respectively, further justifying the high irreversibility of amorphous [238,239].

In addition, D. Guan et al. [240] reported high lithium ion insertion capacity for amorphous TiO_2_ nanotubes. Anodically prepared amorphous TiO_2_ nanotubes possessed high specific capacities of 533 μA hcm^−2^ at 400 μA cm^−2^ along with 77% capacity retention after 50 cycles. This again can be attributed to the high lithium diffusion coefficient and the structural stability of amorphous. Meanwhile, anatase and mixed anatase-rutile phase showed comparatively inferior behavior; the anatase in particular showed crystal defects. Applied voltage contributes to nanostructure morphology, and it also works as an indicator of determining electrochemical behavior. Nanotube array film formed via sequential anodization at 20 V and annealing has good cyclability but a low initial capacity of 180 mAhcm^−3^ attributed to the narrow tube opening. This can be overcome with increases in voltage where 40 V and 60 V showed better intercalation capacity. However, much higher voltage (80 V and 100 V) eroded the tube wall with collapsed morphology and exhibited poor cyclability [241]. Highly ordered nanotubes always make simple pathways for the lithium intercalation mechanism and are associated with better reversible capacities if the nanotubes have thicker walls. Wei et al. [118] synthesized a simple tubular nanostructure of 9 μm length followed by two-step annealing that was observed to provide a high rate capacity of 96 mA hg^−1^ (0.24 mA hcm^−2^) at 5 C rate. Cycling for 500 times at 0.1 C rate showed excellent performance with only 6% initial capacity loss because the stable architecture absorbed the volume strains during intercalation/deintercalation.

It is noted that recently, another promising rechargeable battery, all solid-state LIBs (ASSLIBs), are widely investigated by researchers. ASSLIBs are considered a future energy storage for electric vehicles [242]. This type of battery is not yet available commercially, but it will soon be used in electric vehicles [243]. To the best of our knowledge, there are few research works on anode materials for ASSLIBs. However, Sugiawati et al. [244] reported self-organized TiO_2_ nanotubes as anodes for ASSLIBs. It is noted that this electrode exhibits a high first-cycle Coulombic efficiency of 96.8% with a capacity retention of 97.4% after 50 cycles. In addition, the TiO_2_ nanotubes deliver a stable discharge capacity of 119 mA hg^−1^ at a current rate of C/10. The enhanced electrochemical performance was attributed to the large surface area between the nanotubes and the gel polymer electrolyte, which provides a robust, high-quality electrode–electrolyte interface for long charge–discharge cycles. Chen et al. [245] reported on a TiO_2_ nanofiber-modified lithium metal composite as an anode for solid-state lithium batteries. The solid-state Li-TiO_2_ cell upgrades the critical current density to 2.2 mA cm^−2^ and exhibits stable cycling over 550 h. The enhanced electrochemical performance was attributed to the improved interfacial contact between the garnet electrolyte and the lithium metal anode via the TiO_2_ nanofibers. 

## 8. Conclusions and Outlooks

Extensive studies have been conducted to explore the fabrications and modifications of TiO_2_-based nanostructured anode materials in order to boost the electrochemical performance of LIBs. Nanostructured TiO_2_ displays enhanced specific capacities and improved cyclic performance and rate capacities. The novel knowledge this paper contributes to the literature on this state-of-art energy storage technology is that it gives details on the improved ionic and electronic conductivity of TiO_2_ for better lithium intercalation and the improved specific capacities to fulfill the demands of anodes for high-performance LIBs. The electrochemical performance mainly depends on the size and morphology of the nanostructures. In addition, the phase crystallinity is a great factor to influence the electrochemical performances. Wide variety in dimensions, crystal size, phase, and morphology can be attained by using different preparation techniques. However, an optimum set of size, morphology, crystal phase, and proper preparation technique has been proven effective for improving the diffusion kinetics and electrolyte decomposition, which results in high Coulombic efficiency. High power and energy density requirements are more prominent in modern applications, which are possible with TiO_2_ with high lithium storage capacity and high electronic conductivity. Moreover, the ease of manufacturing, green technology, and manageable cost are facts to be highly considered. In this review article, anodic oxidation, a simple method of TiO_2_ fabrication has been explicitly addressed. The formation mechanisms affecting the nanostructured morphology are controlled by anodizing parameters such as voltage, electrolyte composition, pH, time, and and current density. The hierarchical morphologies of nanostructures of an enormous number of pores and high specific surface areas as well as exposed energy facets that all converge to produce enhanced electrochemical efficiency. It is noted that nanomaterials of TiO_2_ with high specific surface area, optimized surface morphology, and controlled facet show enhanced electron and lithium-ion diffusion rates both within and on the surfaces of these materials. Meanwhile, electrochemical self-doping, annealing, doping with foreign materials, and mixing with other transition materials actively accelerate the enhancement process due to the improvement in the ionic and electronic conductivity. 

It is noted that high energy-density, high cycle-life, and high-efficiency batteries will continue to be the norm in the future expansion of lithium batteries. TiO_2_-based materials could be the future anode materials of LIBs due to their exclusive properties such as fast lithium-ion diffusion, low cost, environmentally friendliness, and good safety. However, these materials suffer from low capacity, low electrical conductivity, poor rate capacity, and lack of scalable synthesis process. Hence, a multidisciplinary effort in this field is necessary in order to use TiO_2_-based materials as effective anodes in commercial LIBs. First, the specific capacity of TiO_2_-based anode materials exhibits ≤400 mA hg^−1^ due to the intercalation mechanism, which limits the applicability of this material high energy-density anode materials. In addition, the high current rate charge and discharge cycling are desirable for application in electric vehicle propulsion. Second, a comprehensive study is necessary to understand the lithiation and delithiation process of this material from the thermodynamics and kinetics points of view. This study will guide future designs of the proper nanostructure for this material. Third, in the quest to use TiO_2_ in practical applications such as anodes for LIBs, there is tremendous variation in the electrochemical performances of lab-produced experimental products. In addition, the fabrication process of the various nanostructures varies so greatly that their electrochemical performances cannot be compared. Hence, it is necessary to find a state-of-art fabrication technology that will fulfill the requirements. 

## Figures and Tables

**Figure 1 nanomaterials-12-02034-f001:**
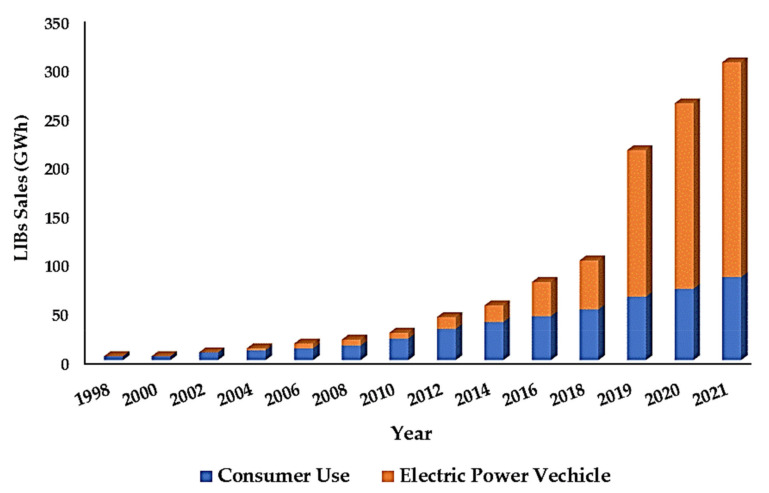
The increasing demand for LIBs.

**Figure 2 nanomaterials-12-02034-f002:**
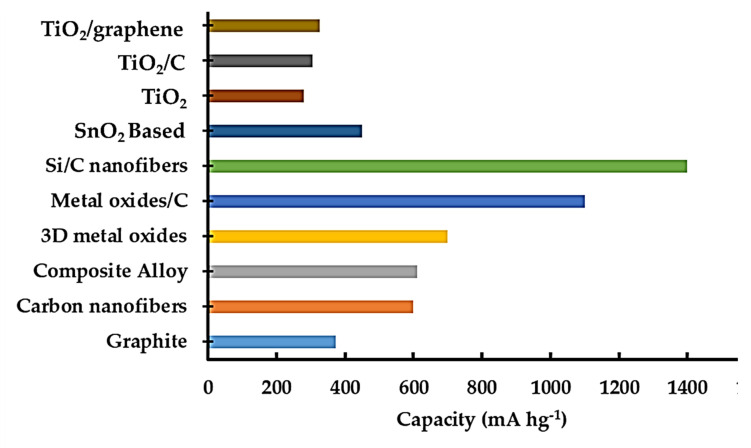
The reversible capacities of different LIB anode materials.

**Figure 3 nanomaterials-12-02034-f003:**
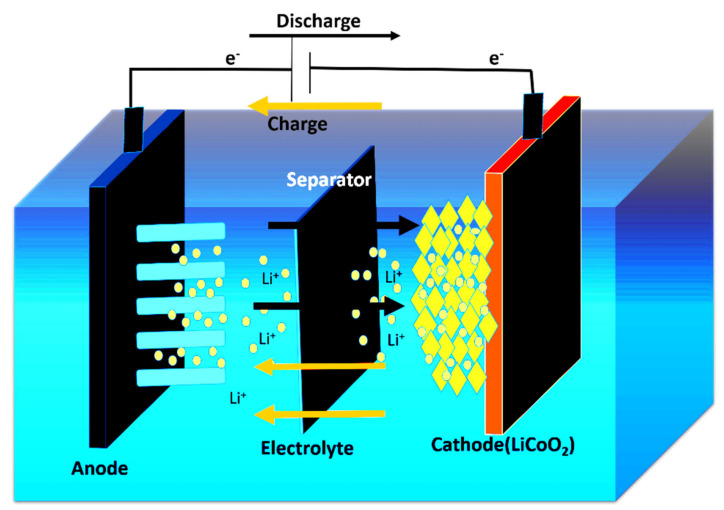
A skeleton of an LIB showing the Li^+^ intercalation/deintercalation mechanism.

**Figure 4 nanomaterials-12-02034-f004:**
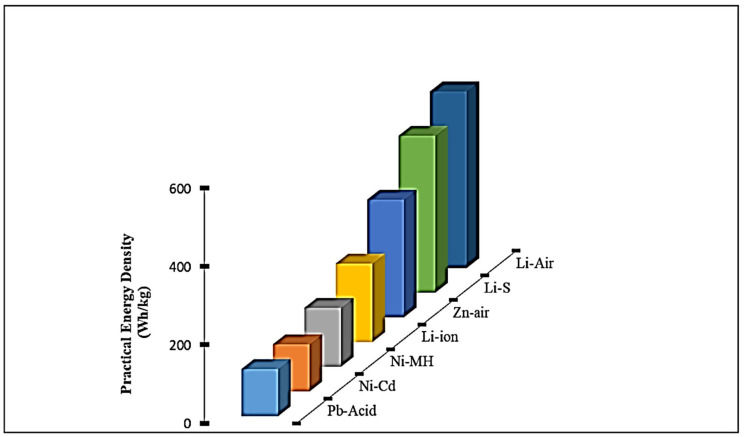
Different types of rechargeable batteries with respect to energy density (Wh kg^−1^).

**Figure 5 nanomaterials-12-02034-f005:**
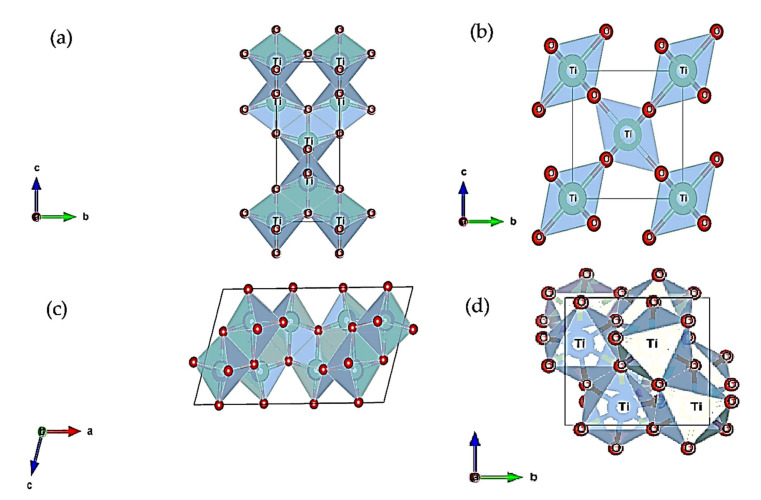
The crystal structures: (**a**) anatase, (**b**) rutile, (**c**) TiO_2_-B, and (**d**) brookite.

**Figure 6 nanomaterials-12-02034-f006:**
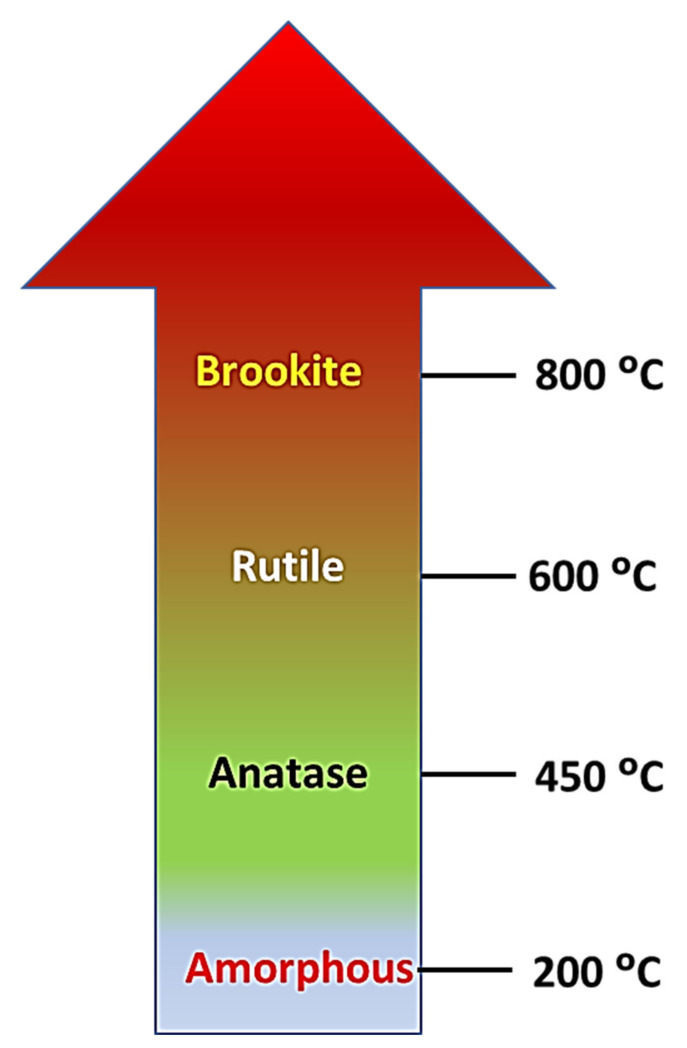
The TiO_2_ crystal phase dependency on annealing temperatures.

**Figure 7 nanomaterials-12-02034-f007:**
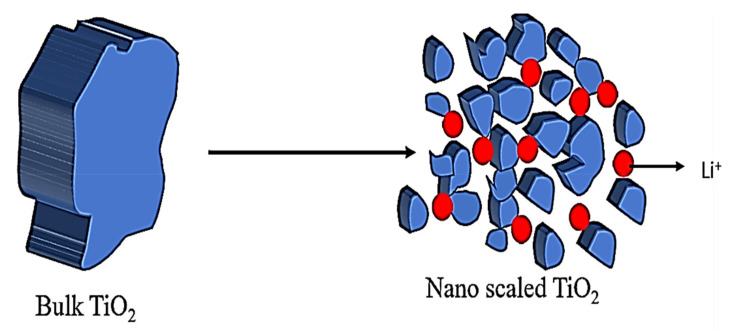
Nano scaling of TiO_2_.

**Figure 8 nanomaterials-12-02034-f008:**
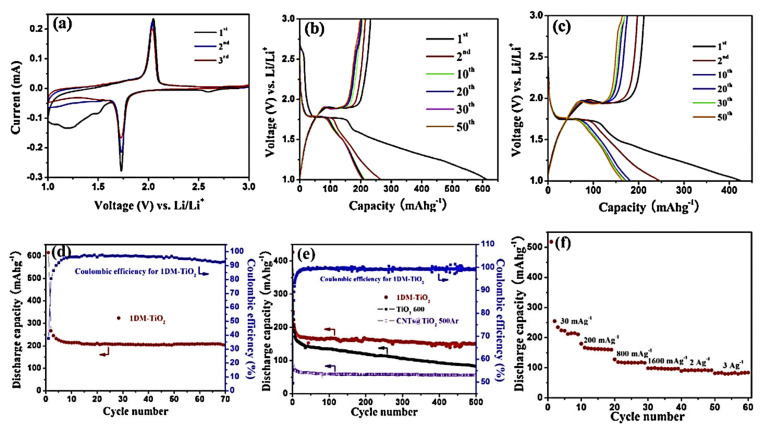
(**a**) The CV diagram. (**b**,**c**) The charge/discharge profiles at 30 mA g^−1^ and 200 mA g^−1^ within 1 V and 3 V. (**d**) The cycling performance and Columbic efficiency of 1DMA-TiO_2_ at 30 mA g^−1^ between 1.0 V and 3.0 V. (**e**) The comparative cycling performance between different samples and Columbic efficiency at 200 mA g^−1^ between 1.0 V and 3.0 V. (**f**) The rate capabilities of 1DM-TiO_2_. Reprinted with permission from Ref. [69]. Copyright 2014 Elsevier.

**Figure 9 nanomaterials-12-02034-f009:**
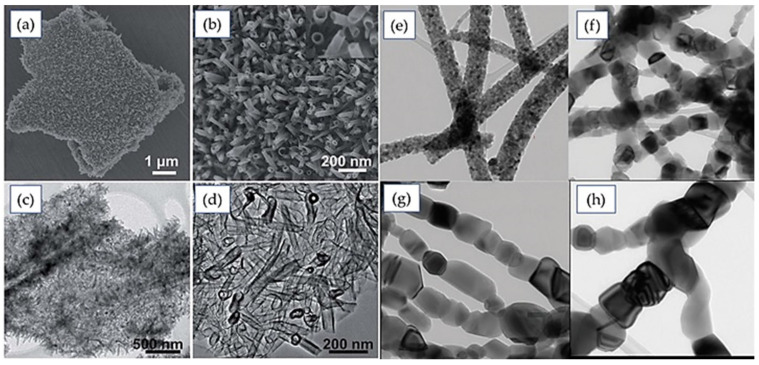
(**a**,**b**) Scanning electron microscope (SEM) images of Gr–TNTs; inset in (**b**) high-magnification image showing TiO_2_ nanotubes on the graphene sheets. (**c**,**d**) Transmission electron microscope images of the Gr–TNTs. Reprinted with permission from Ref. [70]. Copyright 2014 Royal Society of Chemistry. TEM images of carbon-coated TiO_2_ nanofibers (**e**–**h**) calcined in air for 3 h at 400 °C, 600 °C, 800 °C, and 1000 °C, respectively. Reprinted with permission from Ref. [71]. Copyright 2013 Hindawi.

**Figure 10 nanomaterials-12-02034-f010:**
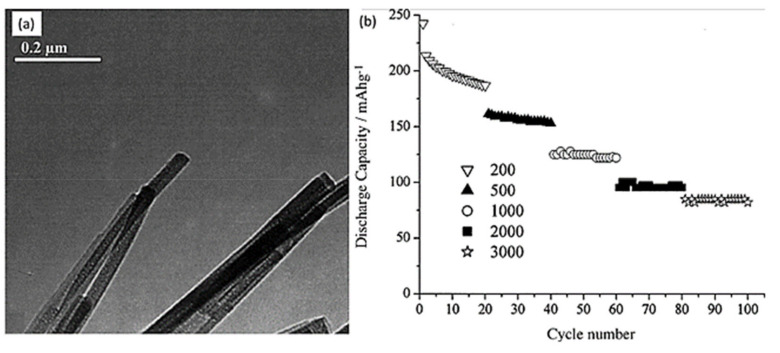
(**a**) TiO_2_-B nanowire TEM image and (**b**) discharge capacities corresponding to cycle number at different current densities. Reprinted with permission from Ref. [72]. Copyright 2005 Elsevier.

**Figure 11 nanomaterials-12-02034-f011:**
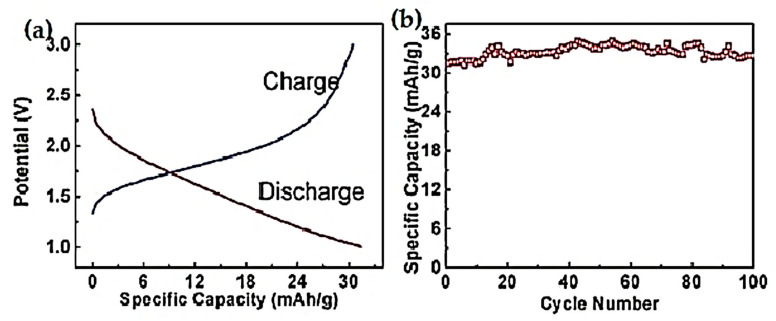
(**a**) Initial charge–discharge profile. (**b**) The corresponding cycle performance of TiO_2_ nanorods as anode of LIBs. Reprinted with permission from Ref. [73]. Copyright 2014 ACS Publications.

**Figure 12 nanomaterials-12-02034-f012:**
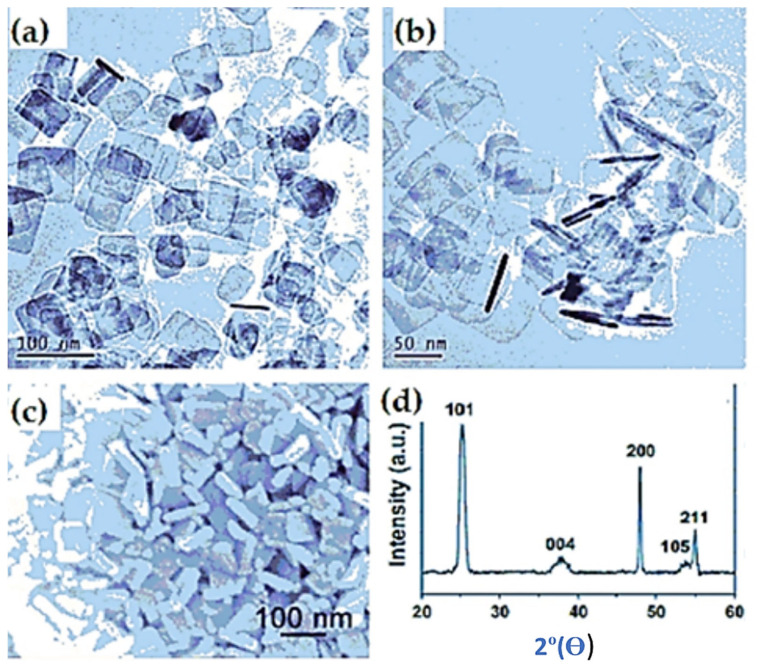
(**a**,**b**) TEM images, (**c**) HRTEM image, and (**d**) XRD pattern of the TiO_2_ nanosheets. Reprinted with permission from Ref. [78]. Copyright 2009 Elsevier.

**Figure 13 nanomaterials-12-02034-f013:**
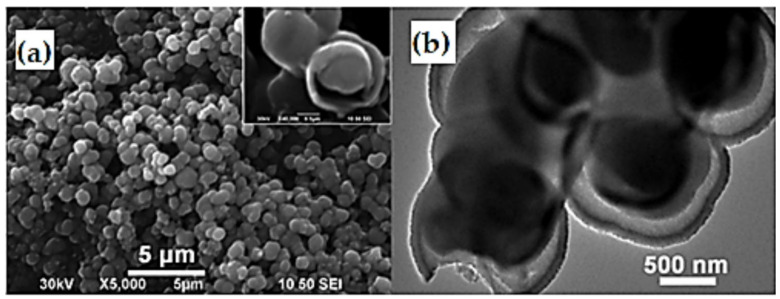
(**a**) SEM and (**b**) TEM images of yolk-shell TiO_2_. Reprinted with permission from Ref. [96]. Copyright 2014 Elsevier.

**Figure 14 nanomaterials-12-02034-f014:**
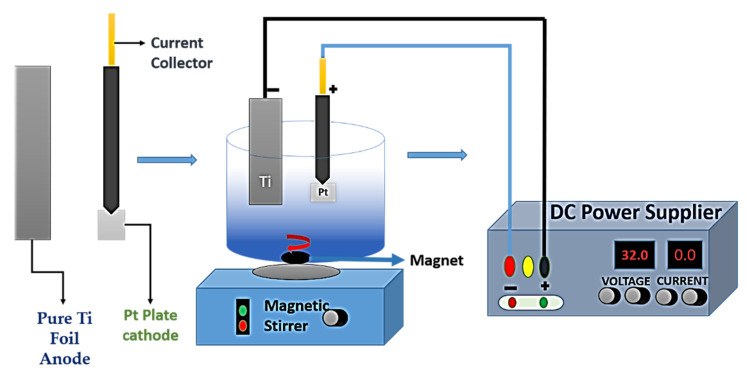
The anodization of pure Ti foil.

**Figure 15 nanomaterials-12-02034-f015:**
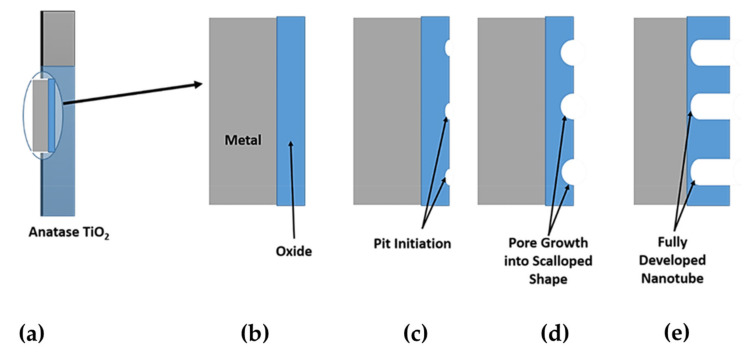
(**a**) Anatase TiO_2_ (large scale). (**b**) The formation of a compact oxide layer. (**c**) The initiation of pits on the oxide layer. (**d**) Pore evolved into a scalloped shape. (**e**) A fully developed nanotube array.

**Figure 16 nanomaterials-12-02034-f016:**
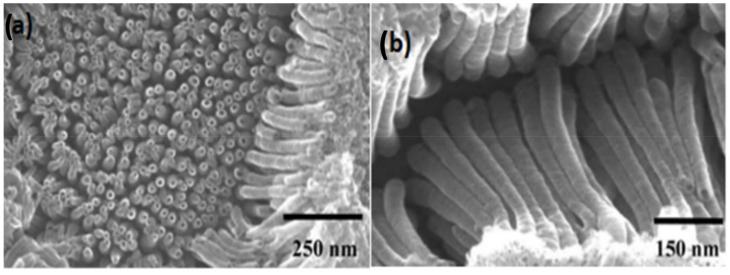
SEM image of TiO_2_ ‘coral reef’ nanotube structures formed at 50 V in CH_3_COOH + 0.5 wt.% NH_4_F solution (**a**) inside and (**b**) outside. Reprinted with permission from Ref. [160]. Copyright 2005 Elsevier.

**Figure 17 nanomaterials-12-02034-f017:**
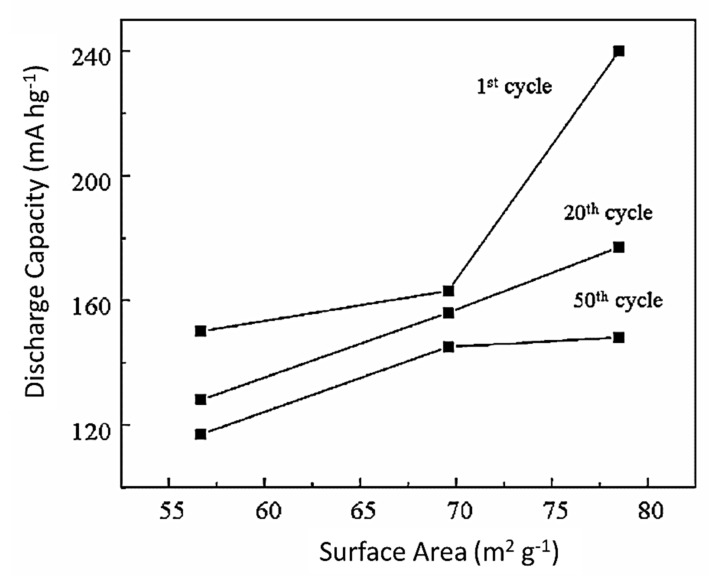
A study of the discharge capacities of anatase TiO_2_ nanotube arrays annealed for three hours at 300 °C, 400 °C, and 500 °C in nitrogen, based on the specific surface areas of the nanotubes. Reprinted with permission from Ref. [187]. Copyright 2010 Elsevier.

**Figure 18 nanomaterials-12-02034-f018:**
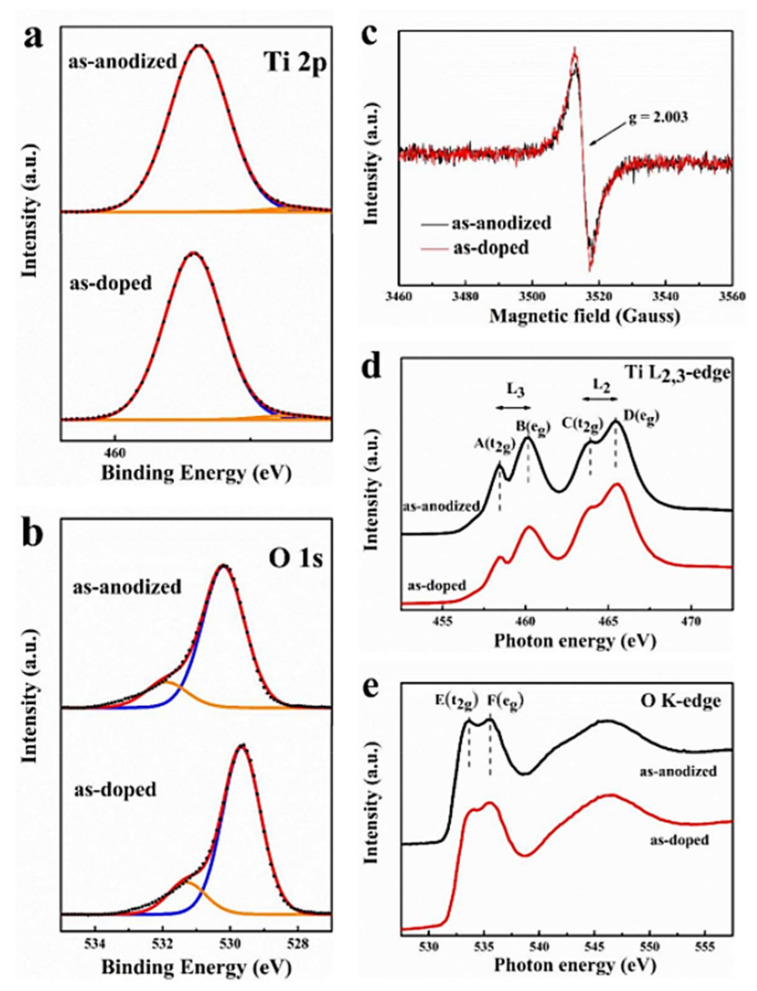
Doped and undoped titanium dioxide nanotubes’ XPS spectra for (**a**) Ti 2p, (**b**) O 1 s, (**c**) EPR, and XANES spectra for (**d**) Ti L2,3-edge XANES and (**e**) O K-edge. Reprinted with permission from Ref. [215]. Copyright 2019 Elsevier.

**Figure 19 nanomaterials-12-02034-f019:**
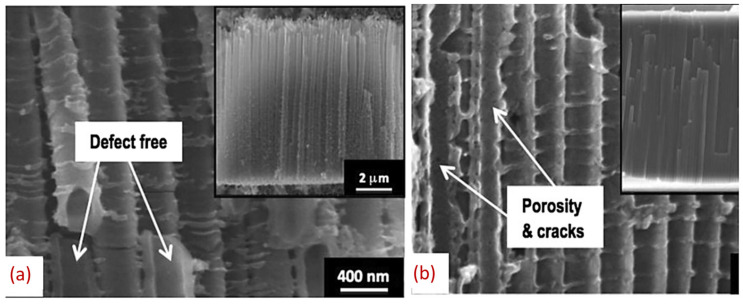
Cross-sections of (**a**) ILNTs and (**b**) AFNTs. Insets in (**a**,**b**): low-magnification side views showing tube lengths. Reprinted with permission from Ref. [234]. Copyright 2014 Elsevier.

**Table 1 nanomaterials-12-02034-t001:** Different TiO_2_ nanostructures and their electrochemical performances.

Preparation Technique	Nanostructures	No of Reversible Cycles	C Rate	Reversible Capacity after 20 Cycles
Solvothermal	Anatase Nanoparticles [106]	100	0.2C	196 mAhg^−1^
	Anatase/Graphene nanosheets [108]	120	1C	161 mAhg^−1^
	C-coated mesoporous			
	TiO_2_@graphenenanosheets [109]	100	0.2 Ag^−1^	111 mAhg^−1^
	Hierarchical nanosheets [107]	700	1C	225 mAhg^−1^
Hydrothermal	TiO_2_@C hollow spheres [111]	300	2C	170 mAhg^−1^
	C-doped TiO_2_ nanowires [110]	1000	0.1C	306 mAhg^−1^
	TiO_2_ nanotube@graphene [112]	50	10 mAg^−1^	357 mAhg^−1^
Hydrolysis	TiO_2_ nanoparticles@graphene [120]	400	2C	70 mAhg^−1^
	Core-shell CNTs/TiO_2_ [121]	100	5 Ag^−1^	240 mAhg^−1^
	Anatase Nanorods [114]	30	5 Ag^−1^	206 mAhg^−1^
	Anatase Nanowires [113]	40	50 mAg^−1^	280 mAhg^−1^
	TiO_2_/graphene [122]	100	140 mAg^−1^	230 mAhg^−1^
Electrospinning	TiO_2_ Nanofibers [71]	50	0.3C	170 mAhg^−1^
	TiO_2_/C hollow nanofibers [115]	100	100 mAg^−1^	229 mAhg^−1^
	Graphene wrapped TiO_2_ nanofibers [116]	35	0.1 mAg^−1^	200 mAhg^−1^
Template Assisted	TiO_2_ nanoporous hollow spheres [123]	50	33.5 mAg^−1^	230 mAhg^−1^
	Mesoporous TiO_2_ nanoparticles [124]	30	0.2C	268 mAhg^−1^
Anodization	TiO_2_ Nanotubes [117]	1000	0.1C	140 mAhg^−1^
	3D Free Standing TiO_2_ nanotubes [118]	500	0.05 mAcm^−2^	184 mAhg^−1^
	SnO_2_/TiO_2_ Nanotubes [125]	50	0.1C	250 mAhg^−1^
	Ti-Mn-O Nanotubes [126]	30	175 mAg^−1^	474 mAhg^−1^
	CoO/TiO_2_ Nanotubes [119]	100	10 μAhcm^−2^	600 μAhcm^−1^
	TiO_2_-SnO_2_ Nanotubes [127]	400	504 μAhcm^−2^	405 μAhcm^−1^
	CoO/TiO_2_ Nanotubes [127]	90	50 μAhcm^−2^	450 μAhcm^−1^
Thermal Treatment	Fe_2_O_3_ nanorods-TiO_2_ [128]	1000	1 Ag^−1^	860 mAhg^−1^
Commercial	Rutile nanoparticles [103]	20	0.05 Ag^−1^	207 mAhg^−1^

**Table 2 nanomaterials-12-02034-t002:** Various dimensions of anodic nanotubes found by altering the anodization parameters.

Voltage (V)	Time	Electrolyte	Average Diameter	Length	
20	60 min	1 M Na_2_SO_4_ +			[144]
0.5 wt.% NH_4_ +	50–80 nm	Negligible
1 wt.% NH_4_	20–30 nm	Negligible
3 wt.% NH_4_	75–100 nm	670–730 nm
5 wt.% NH_4_	75–100 nm	650–680 nm
1 M (NH4)2SO_4_ +		
0.5 wt.% NH_4_	80–100 nm	Negligible
1 wt.% NH_4_	90–130 nm	Negligible
3 wt.% NH_4_	110–130 nm	450–480 nm
5 wt.% NH_4_	60–90 nm	300–340 nm
32	120 min	0.5 wt.% NH_4_F +			[145]
1 M (NH_4_)_2_SO_4_ +	
EG 5 vol%	14.7 ± 8.2 nm
10 vol%	12.8 ± 6.8 nm
30 vol%	11 ± 5.5 nm
50 vol%	26.7 ± 13.6 nm
32	120 min	0.4 wt.% NH_4_F +1 M (NH_4_)_2_SO_4_ +EG 5 vol%	15.70 ± 17.70 nm		[173]

18.75 ± 15.40 nm
(Pre heat treated Ti foil)
20	18 h	0.5 wt.% NaF +0.5 M Na_2_SO_4_ +0.5 M H_3_PO_4_ +0.2 M Na_3_C_6_H_5_O_7_ + NaOHwithpH = 1pH = 3pH = 4.2pH = 5			[155]




110 nm	1 μm
110 nm	1.5 μm
110 nm	2.6 μm
110 nm	3 μm
40	70 h	DMSO electrolyte	120 nm		[146]
with	
1 wt% HF	53 μm
2 wt% HF	13 μm
4 wt% HF	53 μm
6 wt% HF	52 μm
60	20 min30 min40 min	EG containing 0.3 wt.% NH_4_F and 2 vol% H_2_O	120 nm	18 μm25.2 μm30.8 μm	[174]
30	3 h	EG containing 0.25 wt.% NH_4_F and 10 wt.% H_2_O	100 nm	4 μm	[175]
20	45 min	EG containing 3 wt.% NH_4_F and 2 vol% H_2_O	40 nm	1.2 μm	[138]
60	3 h	EG containing 0.3 wt.% NH_4_F and 2 vol% H_2_O	120 nm	46 μm	[137]
50	30 min	EG containing NH_4_F and DI water	34 nm	100 nm	[135]
52	44 nm	100 nm
55	53 nm	100 nm
57	58 nm	200 nm
25	2 h	EG containing 0.27 M NH_4_F and 0.2 wt.% H_2_O	58.25 nm	1.04 μm	[136]
30	70.29 nm	3.32 μm
35	91.24 nm	3.68 μm
40	77.33 nm	3.21 μm
45	96.98 nm	4.27 μm
50	119.68 nm	8.35 μm
40	19 h	EG containing 0.27 M NH_4_F, 1.5 wt.% H_2_O, 0.05% HF aqueous solution	75 nm	1.3 μm	[170]

## Data Availability

All data are provided in the manuscript.

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
