# Peer review of "TiO2 as an Anode of High-Performance Lithium-Ion Batteries: A Comprehensive Review towards Practical Application"

_nanomaterials, 2022, doi:10.3390/nano12122034_

Round 1

Reviewer 1 Report

In this review, the authors innovatively summarized the recent research progress and achievements of TiO2 anode materials in Li-ion batteries. A comprehensive summary of nanostructures, synthetic methods, and electrochemical is discussed in detail. Moreover, electrochemical Anodized TiO2 anodes are put forward from different aspects. In my view, this review is timely and interesting and could provide some insights for further developing TiO2 anodes. Thus, this manuscript can be published in Nanomaterials after minor revision. Here are the comments in detail:

(1) In the introduction, I'm not sure if it is an accurate expression that “Chemist John Goodenough and, his research team including Phil Wiseman, Koichi Mizushima and Phil Jones from Oxford University first proposed Lithium-ion battery in the 1970s”. It looks like Stanley Whittingham proposed the first lithium-ion battery in the 1970s. Chemist John Goodenough made great contributions to practical cathodes of lithium-ion batteries.

(2) In Figure 1, is it possible to add the demand for LIBs in the years 2019-2021? Besides, the quality of Figures 21-23 needs to be improved.

(3) In section 7.3, the authors have mentioned that TiO2 a semiconductor material possesses a wide bandgap of 3.2 eV. Do all TiO2 with different crystal phases (anatase, rutile, brookite, ramsdellite, spinel) have the same bandgap value?

(4) Many mesoporous TiO2 anodes for Li-ion batteries with good performance are missing, like 2D TiO2 mesoporous nanosheets (Chemical Engineering Journal, 2021, 420, 129894), 3D hollow TiO2 microspheres (Nano Research 2016, 9, 165), 1D TiO2 nanocrystalline (Crystals 2020, 10, 493) etc.

(5) TiO2 anodes for Li-ion batteries also have many drawbacks compared to other anodes, especially for practical applications. For example, the relatively high operating voltage of TiO2 anodes usually leads to low voltage of Li-ion full cells, which is bad for achieving high energy density. Some comments on the disadvantages and further improvement strategies of TiO2 anodes should be provided in this review. 

Author Response

Reviewer #1

Responses to reviewer’s comments

The authors would like to thank the reviewer for the valuable comments on the paper and have made changes in line with the comments.

In this review, the authors innovatively summarized the recent research progress and achievements of TiO2 anode materials in Li-ion batteries. A comprehensive summary of nanostructures, synthetic methods, and electrochemical is discussed in detail. Moreover, electrochemical Anodized TiO2 anodes are put forward from different aspects. In my view, this review is timely and interesting and could provide some insights for further developing TiO2 anodes. Thus, this manuscript can be published in Nanomaterials after minor revision. Here are the comments in detail:

Comment 1: In the introduction, I'm not sure if it is an accurate expression that “Chemist John Goodenough and, his research team including Phil Wiseman, Koichi Mizushima and Phil Jones from Oxford University first proposed Lithium-ion battery in the 1970s”. It looks like Stanley Whittingham proposed the first lithium-ion battery in the 1970s. Chemist John Goodenough made great contributions to practical cathodes of lithium-ion batteries.

Response: This comment is highly appreciated and the paragraph has been edited and added to paragraph 1, page 1 in Section 1 “Introduction”.

“M. Stanley Whittingham while working for an oil company Exxon, first proposed the idea of rechargeable LIBs in the 1970s. The LIB was composed of metallic lithium as anode and layered titanium disulfide as cathode. Later in the early 1980s, Goodenough flourished the rechargeable LIBs field with layer oxide cathode materials [9].”

Comment 2: In Figure 1, is it possible to add the demand for LIBs in the years 2019-2021? Besides, the quality of Figures 21-23 needs to be improved.

Response: We appreciate this comment and edited the Figure 1 with the demands of LIBs in the years 2019-2021 in Page 2.

However, we have discarded the Figures 21 and 22 of original manuscript due to copyright issues in current version of the manuscript. In addition, we have increased the quality of the Figure 23 (original manuscript) which is now Figure 18.

Comment 3: In section 7.3, the authors have mentioned that TiO2 a semiconductor material possesses a wide bandgap of 3.2 eV. Do all TiO2 with different crystal phases (anatase, rutile, brookite, ramsdellite, spinel) have the same bandgap value?

Response: No, the band gap of anatase, rutile, and Brookite have little difference. We have addressed this comment in paragraph 2 of page 32 as follows;

“TiO2 a semiconductor material possesses wide band gap (rutile ⁓2.98 eV, anatase ⁓3.05 eV, and brookite ⁓3.26 eV) at neutral condition [205].”

Comment 4: Many mesoporous TiO2 anodes for Li-ion batteries with good performance are missing, like 2D TiO2 mesoporous nanosheets (Chemical Engineering Journal, 2021, 420, 129894), 3D hollow TiO2 microspheres (Nano Research 2016, 9, 165), 1D TiO2 nanocrystalline (Crystals 2020, 10, 493) etc.

Response: We appreciate this comment and addressed this comment in pages 14, 15, 16, and 17 in revised manuscript. We have highlighted in red by citing above mentioned all references too.

Page 14

“Nanocomposite of hexagonal MoO3 inlaid with highly crystalline TiO2 nanoparticles have been synthesized by Adam Kubiak et al.[76] following a template assisted microwave method. The optimum molar ratio 5:5 was found for the best outcomes. That nanocomposite as anode of LIBs can sustain 100 cycles with reversible discharge capacity 700 mAhg-1 which is more than the two times of Anatase TiO2. The highly active electrochemical nature of MoO3 is the precursor for these outstanding outcomes. But Anatase TiO2 nanoparticles also has great impacts by making the cyclic operation safer and maintaining the stability of the architecture suppressing the volume fluctuations.

Another TiO2 nanocomposite comprising nanocrystal TiO2 ­and N-doped carbon nanotubes (TiO2@C/N) was used as binder free anode material for LIBs. The anode also showed very benign results. It showed 1054.7 mAhg-1 initial discharge capacity, extremely higher than Anatase TiO2 nanocrystal. The synergistic effects of highly conductive N-doped carbon nanofibers and highly porous TiO2 nanocrystals are the contributing factors for superior cycle stability, high-rate capability and high reversible capacities. Porous TiO2 nanocrystals embedded on carbon nanotubes offered facile channels for charge transfer and shortened the diffusion lengths. Moreover, integrated flexible mechanical structure was offered by N-doped carbon nanotubes [77].

1D nanoribbons of showed higher electrochemical behaviors as anode when compiled as a nanocomposite with SnO2. Reversible discharge capacity 265 mAhg-1 was displayed. The main drawback of SnO2 is rapid capacity degradation due to architectural instability and high-volume fluctuations. While TiO2-(B) nanoribbons are implanted here the mechanical supports are ensured and at the same time higher specific capacity is maintained by SnO2.[78].

Pages 15 and 16

“A scalable template assisted solgel method has been implied to synthesis 2D TiO2 mesoporous nanosheets. The interfacial stain between the template KCl and TiO2 suppressed the crystallization process of TiO2, though the reaction temperature was maintained about 400 ºC. Amorphous TiO2 nanosheets possessed high pseudo capacity, more improved diffusion kinetics and high durability. As a result, it exhibited 103 mAhg-1 reversible capacity at high C rate 6 Ahg-1 even after 1000 cycles[81].

Carbon coated TiO2 nanosheets/reduced graphene oxide were prepared by Shang Jiang et al. through a simple one pot solvothermal process. 10 nm ultrathin nanosheets are homogeneously moored on the reduced graphene sheets. This heterogenous hierarchical structure provides facile pathways for lithium charge transfer and higher cycling stability. As anode this nanocomposite displayed 401 mAhg−1 after 200 cycles at a current density of 100 mAg−1 and 126.5 mAhg−1 at a current density of 2500 mAg−1. These results confirm better rate capability and cyclability than pure Anatase TiO2 nanosheets[82].

2D nanosheets of TiO2 were used to make composites with 0D SnO2 nanoparticles in the Qinghua Tian et al.[83] study. A simple morphology-maintained phase transformation process was applied. Enhanced lithium storage capability like high specific capacity of 758 and 474 mAhg−1 at 200 and even 1000 mAg−1 after 390 and 650 cycles, were showed respectively. The combined effects of outer carbon layer and inner robust structure of 2D TiO2 nanosheets and 0D SnO2 nanoparticles are the main attributes for better electrochemical kinetics, higher architectural stability and enhanced lithium-ion storage capacity.

Dendrite formation is a serious issue in battery cycling. Moreover, high volume fluctuations cause rapid capacity fall. To suppress these issues, another approach has been adopted by synthesizing a novel 2D TiO2 nanosheets combined with graphene and coated by 2D black phosphorous nanosheets (BPN). This heterostructure promotes rapid interlayer charge transfer paths and higher lithium storage capability. Both BPN and TiO2 nanosheets possess highly conductive nature due to their advantageous nanoarchitecture. In consequence, superior rate capability (271.1 mAhg-1 at 5 Ag-1), enhanced cyclability (502 mAhg-1 after 180cycles) and highly improved storage capacity (initial discharge 1336.1 mAhg-1 at 0.2 Ag-1) were observed within 1-3V potential window as anode. These outcomes are definitely far more better than any pure nanostructured Anatase TiO2 [84].

Jorge Lopez et al. [85] prepared a nanocomposite of TiO2/C where, unique 2D nanoparticles were used with 2D carbon nanofibers. The heterogeneous nanostructure showed improved rate capability (200 mAhg-1 at 500 mAg-1 current density), good capacity retention (⁓99%), high initial discharge capacity (683 mAhg-1) and better cycle stability (290 mAhg-1 after 100 cycles at 100 mAhg-1 current density). The layered TiO2 nanoparticles allowed fast charge transfer and improved the diffusion kinetics. Moreover, the highly conductive 2D carbon nanofibers accelerated the ion conductivity and resulted in higher electrochemical performances.”

Page 16

“Another hetero nanostructure combined of ultrathin 2D mesoporous TiO2 and reduced graphene showed very good cyclability and rate capability (245 mAhg-1 at 1Ag-1 after 1000 cycles). The volume fluctuations during lithium insertion and extraction were very slight due to flexible mechanical support from 2D mesoporous TiO2. In addition, faster kinetics and short diffusion rate of ions were confirmed by this stacked layer structure [87].

r-GO@TiO2(B)@Mn3O4 a multifunctional, multi dimensionally ordered yolk-membrane-shell hybrid nanostructure where TiO2(B) nanosheets acted as major active materials. Most the capacity contributions came from TiO2(B) as well as safety and durability. Meanwhile, auxiliary active material Mn3O4 possessed high initial discharge capacity 935 mAhg-1. r-GO nanosheets acted like ion host and absorbed mechanical stress of TiO2(B). Aggregation of Mn3O4 was suppressed by TiO2(B) which in consequence caused strong lithium storage and delivering capacity (662 mAhg-1 at 500mAg-1 after 500 charge discharge cycles) [88].”

Page 17

“Bio Chen et al. [92] prepared another sandwich like nanostructure where carbon coated ultrathin TiO2 nanosheets (⁓14 nm) were endowed with defect rich MoS2 from both sides via intimate interfacial contact. The defects rich few layers of MoS2 and the ultrathin TiO2 nanosheets compensate the volume fluctuations during intercalation to a great extent. In consequence, higher rate capability (792.3 mAhg-1 at 100 mAg-1) and better cycle performance (805.3 mAhg-1 at 0.1 Ag-1 after 100 cycles) were appeared for this anode.”

Comment 5: TiO2 anodes for Li-ion batteries also have many drawbacks compared to other anodes, especially for practical applications. For example, the relatively high operating voltage of TiO2 anodes usually leads to low voltage of Li-ion full cells, which is bad for achieving high energy density. Some comments on the disadvantages and further improvement strategies of TiO2 anodes should be provided in this review. 

Response: We highly appreciate this comment and addressed this comment in section ‘8. Conclusion and Outlooks’ as follows;

“TiO2-based materials could be the future anode materials of LIBs due to their exclusive properties such as fast lithium ion diffusion, low cost, environmentally friendly, and good safety. In contrast, these materials suffer from low capacity, low electrical conductivity, poor rate capability, and lack of scalable synthesis process. Hence, a multidisciplinary effort in this field is necessary in order to use TiO2-based materials as effective anodes in commercial LIBs. First, the specific capacity of TiO2-based anode materials exhibits ≤ 400 mA h g-1 due to the intercalation mechanism which is limited to consider as high energy density anode materials. In addition, the high current rate charge and discharge cycling is desirable when their application is considered for electric vehicle propulsion. Second, comprehensive study is necessary to understand the lithiation and delithiation process of this material from thermodynamics and kinetics point of view. This study will guide to design the proper nanostructure design of this material. Third, quest to practical applications as anode of LIBs, there is huge gap of electrochemical performances of lab experimental results and reality. In addition, the fabrication process of the various nanostructures varies so high which cannot be compared with respect to electrochemical performances. Hence, it is necessary to find the state-of-art fabrication technology which will allow to fulfill the requirements. ”

Reviewer 2 Report

This is a good review article and worth to be published in nanomaterials. Please propose some characterization methods to better shed light on the TiO2 anodes. This may be suitable for the last section. I also recommend to add some sentenced about the potential applications of TiO2 in All-Solid-State batteries. 

Author Response

Reviewer #2

Responses to reviewer’s comments

The authors would like to thank the reviewer for the valuable comments on the paper and have made changes in line with the comments.

Comment: This is a good review article and worth to be published in nanomaterials. Please propose some characterization methods to better shed light on the TiO2 anodes. This may be suitable for the last section. I also recommend to add some sentenced about the potential applications of TiO2 in All-Solid-State batteries. 

Response: This comment is highly appreciated and we tried in our limit to explain the electrochemical performances of various TiO2 as electrodes of LIBs. However, the detail explanation of all characterization methods is not our scope in this present review.

In addition, we have addressed the potential application of TiO2 in All-Solid-State batteries in section 7.3.4 in page 39 by adding following paragraph. 

“It is noted that recently another promising rechargeable All Solid-State LIBs (ASSLIBs) is widely investigated by the researchers. ASSLIBs is considered as future energy storage for electric vehicles [250]. This type of battery still has not been commercially used, however, it will soon be used in electric vehicles [251]. To our best knowledge, there are not so many research works on anode materials of ASSLIBs. However, Sugiawati et al. [252] reported self-organized TiO2 nanotubes as anode of ASSLIBs. It is noted that this electrode exhibits a high first-cycle Coulombic efficiency of 96.8% with a capacity retention of 97.4% after 50 cycles. In addition, TiO2 nanotubes delivers a stable discharge capacity of 119 mAh g−1 at a current rate of C/10. The enhanced electrochemical performance was attributed to the large surface area between the nanotubes and the gel polymer electrolyte, which provides a robust and a high quality electrode–electrolyte interface for long charge-discharge cycles. Chen et al. [253] reported TiO2 nanofiber-modified lithium metal composite as anode for solid-state Lithium Batteries. The solid state Li-TiO2 cell upgrades the critical current density to 2.2 mA cm–2 and exhibits stable cycling over 550 h. The enhance electrochemical performance was attributed due to the improved interfacial contact between the garnet electrolyte and the lithium metal anode via TiO2 nanofibers. ”

Reviewer 3 Report

This review work is focused on the use of TiO2 as Anode of High Performance Lithium-ion Batteries.

This is an interesting work, taking into account the necessary references.

I feel this work could be published in its present form.

Just a few typos should be corrected.

Author Response

Reviewer #3

Responses to reviewer’s comments

The authors would like to thank  the reviewer for the valuable comments on the paper and have made changes in line with the comments.

Comment: This review work is focused on the use of TiO2 as Anode of High Performance Lithium-ion Batteries. This is an interesting work, taking into account the necessary references. I feel this work could be published in its present form. Just a few typos should be corrected.

Response: We appreciate this comment and edited the manuscript accordingly.
